# Phase transitions of multivalent proteins can promote clustering of membrane receptors

**Sudeep Banjade, Michael K Rosen\***

Department of Biophysics, Howard Hughes Medical Institute, University of Texas Southwestern Medical Center, Dallas, United States

**Abstract** Clustering of proteins into micrometer-sized structures at membranes is observed in many signaling pathways. Most models of clustering are specific to particular systems, and relationships between physical properties of the clusters and their molecular components are not well understood. We report biochemical reconstitution on supported lipid bilayers of protein clusters containing the adhesion receptor Nephrin and its cytoplasmic partners, Nck and N-WASP. With Nephrin attached to the bilayer, multivalent interactions enable these proteins to polymerize on the membrane surface and undergo two-dimensional phase separation, producing micrometer-sized clusters. Dynamics and thermodynamics of the clusters are modulated by the valencies and affinities of the interacting species. In the presence of the Arp2/3 complex, the clusters assemble actin filaments, suggesting that clustering of regulatory factors could promote local actin assembly at membranes. Interactions between multivalent proteins could be a general mechanism for cytoplasmic adaptor proteins to organize membrane receptors into micrometer-scale signaling zones.

## Introduction

Numerous membrane proteins have been observed to organize into supramolecular clusters upon extracellular ligand binding and/or cell–cell adhesion. Examples include cadherins (*Yap et al., 1997*), Eph receptors (*Nikolov et al., 2013*; *Seiradake et al., 2013*), immune receptors (*Goldstein and Perelson, 1984*), apoptotic signaling receptors (*Henkler et al., 2005*), chemotaxis receptors (*Li et al., 2011*), GPI anchored proteins (*Varma and Mayor, 1998*) and components of T cell signaling pathways (*Balagopalan et al., 2013*). A variety of mechanisms have been proposed to account for this higher-order organization. The extracellular domains of cadherins and Eph receptors have been postulated to interact laterally in homotypic fashion within the plasma membrane to produce large-scale assemblies at sites of cell–cell adhesion (*Himanen et al., 2007*; *Wu et al., 2011*; *Seiradake et al., 2013*). Modeling studies have suggested that binding of divalent antibodies to the extracellular domain of trivalent Fcε receptors could lead to large networks, which could account for Fcε receptor puncta observed in cells (*Goldstein and Perelson, 1984*). An analogous mechanism has been proposed for intracellular inter-actions of the oligomeric receptor, Fas, with its oligomeric adaptor protein FADD (*Scott et al., 2009*; *Wang et al., 2010*; *Wu, 2013*) to produce clusters of the receptors hundreds of nanometers in size (*Siegel et al., 2004*). Similarly, dimeric bacterial chemoreceptors such as Tsr are linked together by their downstream partners CheA and CheW, forming trimers of dimers resulting in a highly ordered and conserved hexagonal array that is suggested to be the basic unit of polar clusters (*Briegel et al., 2012*, *2014*). GPI-anchored proteins and lipid-anchored Ras have been shown to organize into dynamic clusters of 4–7 molecules through transient interactions with lipids and the cortical actin–myosin network (*Plowman et al., 2005*; *Goswami et al., 2008*). Such clusters of GPI-anchored proteins are also believed to play an important role in creation of dynamic nanometer scale cholesterol-rich lipid domains, which further contribute to organization of the plasma membrane (*Sharma et al., 2004*;

**\*For correspondence:** michael.rosen@utsouthwestern.edu

**Competing interests:** The authors declare that no competing interests exist.

**eLife digest** The membrane that surrounds a cell is made up of a mixture of lipid molecules and proteins. Membrane proteins perform a wide range of roles, including transmitting signals into, and out of, cells and helping neighboring cells to stick together. To perform these tasks, these proteins commonly need to bind to other molecules—collectively known as ligands—that are found either inside or outside the cell.

Membrane proteins are able to move around within the membrane, and in many systems, ligand binding causes the membrane proteins to cluster together. Although this clustering has been seen in many different systems, no general principles that describe how clustering occurs had been found.

Now, Banjade and Rosen have constructed an artificial cell membrane to investigate the clustering of a membrane protein called Nephrin, which is essential for kidneys to function correctly. When it is activated, Nephrin interacts with protein ligands called Nck and N-WASP that are found inside cells and helps filaments of a protein called actin to form. These filaments perform a number of roles including enabling cells to adhere to each other and to move.

In Banjade and Rosen's artificial system, when a critical concentration of ligands was exceeded, clusters of Nephrin, Nck and N-WASP suddenly formed. This suggests that the clusters form through a physical process known as 'phase separation'. Banjade and Rosen found that this critical concentration depends on how strongly the proteins interact and the number of sites they possess to bind each other.

Within the clusters, the three proteins formed large polymer chains. The clusters were mobile and, over time, small clusters coalesced into larger clusters. Even though the clusters persisted for hours, individual proteins did not stay in a given cluster for long and instead continuously exchanged back-and-forth between the cluster and its surroundings.

When actin and another protein complex that interacts with N-WASP were added to the artificial membrane system, actin filaments began to form at the protein clusters. Banjade and Rosen suggest that such clusters act as 'signaling zones' that coordinate the construction of the actin filaments. Regions that are also found in many other signaling proteins mediate the interactions between Nephrin, Nck and N-WASP. Banjade and Rosen therefore suggest that phase separation and protein polymer formation could explain how many different types of membrane proteins form clusters.

*Lingwood and Simons, 2010*; *Gowrishankar et al., 2012*). Finally, data suggest that clustering of T cell receptors may arise in part from size differences, and consequent steric occlusion, between the extracellular domains of different membrane proteins found at contacts between T cells and antigen presenting cells (*James and Vale, 2012*).

These models have proven powerful in describing the individual systems above. However, for several reasons most of them are difficult to generalize in a predictive manner to new systems. First, the models hinge on molecular interactions that are specific to the individual systems and that are not readily apparent from protein sequence features alone. Thus, with the exception of GPI-anchored proteins, which likely behave similarly as a group, in the absence of a fairly detailed physical characterization it is difficult to predict whether any new protein/system is likely to produce membrane clusters, and if so, which clustering models are most appropriate. Additionally, with the exception of the nanometer scale clusters of GPI-anchored proteins and lipid-modified Ras, the physical properties of most receptor clusters have not been extensively characterized. Clustering models have derived in many cases from molecular packing in crystal lattices and have been analyzed largely through cellular studies showing qualitative consistency with structural studies and theoretical analyses. But in general, the physical properties of the clusters (e.g., their thermodynamic and kinetic properties) have not been correlated to the physical parameters of the molecules that compose them nor have the key molecular properties that influence cluster properties been identified. This shortcoming arises partly because the models have not been examined through in vitro biochemical reconstitution, where the parameters of the system can be tightly controlled and the physical properties of the clusters can be analyzed in detail. Finally, the functional consequences of macroscopic clustering (as distinct from association to create defined oligomers–dimers, trimers, etc) are not well understood. But it is notable that many clustered receptors signal to the actin cytoskeleton, and that many of their downstream targets, such

as actin nucleation promoting factors in the WASP family, are also known to form micrometer sized clusters at the plasma membrane (*Yamaguchi et al., 2005*; *Weiner et al., 2007*; *Gomez and Billadeau, 2009*). These observations suggest that one function of receptor clustering may be to control the localization, structure, and/or dynamics of actin filament networks.

We recently demonstrated that interactions between multivalent proteins and their multivalent ligands can lead to macroscopic phase separation. This occurs concomitant with assembly of the proteins into large polymers, through a sol–gel transition, as observed in many other multivalent systems in polymer science (*Li et al., 2012*). In three-dimensional solution, this process produces phase separated protein polymers that organize into dynamic micron sized liquid droplets. These droplets are formed in a sharp transition as protein concentration in solution is increased. The critical concentration for droplet formation depends on valency and affinity of interacting species, and the proteins are highly concentrated within the droplets. We have studied this phenomenon in a variety of model multivalent systems, involving both protein–protein and protein–RNA interactions, and also in an actin regulatory signaling pathway involving the adhesion receptor, Nephrin, and its intracellular targets Nck and N-WASP (*Jones et al., 2006*). In the latter, phase separation can be controlled by multivalent phosphorylation of Nephrin and results in enhanced signaling activity of N-WASP.

These previous studies were performed in three-dimensional solution. But in vivo Nephrin is an integral membrane protein; therefore its cytoplasmic tail is attached to membranes (*Welsh and Saleem, 2010*). The behavior of multivalent–multivalent interaction systems in such a two-dimensional arrangement remained unresolved. In this study, we show that multivalency-induced polymerization and phase separation can also occur in two-dimensional systems, generating micrometer-size protein clusters at membranes. When phosphorylated Nephrin is attached to supported lipid bilayers of DOPC, addition of Nck and N-WASP induce formation of micron-sized concentrated puncta containing all three proteins. Puncta form abruptly when a critical concentration of Nck/N-WASP is reached and are highly dynamic. The critical concentration is appreciably lower for two-dimensional puncta formation than for three-dimensional droplet formation, and it depends on the phosphotyrosine and SH3 domain valencies of p-Nephrin and Nck, respectively, and also on the affinity of the Nck SH2 domain for p-Nephrin. These data suggest that puncta formation is driven by polymerization of the proteins in a plane adjacent to the membrane. In the presence of actin and the N-WASP target, the Arp2/3 complex, puncta formation causes focal actin assembly. Our biochemical approach has allowed us to control the clustering process and discover key parameters that control puncta formation. Our study demonstrates that specific protein–protein interactions result in the formation of macroscopic clusters without the necessity of lipid segregation or actin–myosin assembly. This clustering can be defined as phase separation of proteins on the surface of a membrane. Our observations here and previously (*Li et al., 2012*) suggest that polymerization and phase separation of multivalent macromolecules may represent a general mechanism to produce two- and three-dimensional dynamic and highly concentrated micron-sized structures in cells.

## Results

### Membrane-bound p-Nephrin clusters through a phase transition upon addition of Nck and N-WASP

Nephrin is a transmembrane protein expressed in the foot processes of kidney podocyte cells, where its extracellular domain is a critical component of the slit diaphragm, the final element of the kidney's glomerular filtration barrier (*Welsh and Saleem, 2010*). The integrity of the slit diaphragm requires intracellular assembly of actin filaments downstream of the Nephrin cytoplasmic tail (*Jones et al., 2006*). When Nephrin is crosslinked by antibodies, its cytoplasmic tail can be phosphorylated by the Src family kinase, Fyn (*Jones et al., 2006*; *Verma et al., 2006*). Three phosphotyrosines (pTyrs) in the tail bind the SH2 domain of the Nck adaptor protein, which in turn uses its three SH3 domains to bind multiple proline-rich motifs (PRMs) in the actin regulatory protein, N-WASP. N-WASP then recruits and promotes activation of the Arp2/3 complex, which generates branched actin filament networks through nucleating new actin polymers. Mutations that disrupt this pathway in humans and mice result in disorganization of the slit diaphragm and defects in the glomerular filter that cause proteinuria (*Jones et al., 2006*, *2009*).

We previously reported that mixing Nck, N-WASP, and the phosphorylated cytoplasmic tail of Nephrin in solution produced phase separated liquid droplets (*Li et al., 2012*). This observation suggested that if the Nephrin tail was attached to a membrane, as it is in vivo, Nck and N-WASP might

induce it to condense into membrane clusters (*Figure 1A*). To test this hypothesis, we began by generating the triply phosphorylated cytoplasmic tail of Nephrin (amino acids 1174–1223, phosphorylated at Tyr1176, Tyr1193, and Tyr1217, and mutated from Tyr to Phe at residues 1183 and 1210, sites not believed to bind Nck (*Jones et al., 2006*; *Verma et al., 2006*); called p-Nephrin hereafter). The construct contained a His$_8$ tag at its N-terminus, followed by a (Gly-Gly-Ser)$_5$ linker containing a cysteine, which was covalently labeled with maleimide Alexa488 fluorophore. We attached p-Nephrin to supported bilayers of DOPC lipid, doped with 1% of a nickel-chelating lipid (Ni$^{2+}$-NTA-DOGS). Through this approach we could control and quantify the surface density of p-Nephrin as detailed in the 'Materials and methods' section (*Galush et al., 2008*).

Membrane-bound p-Nephrin is homogeneous and fluid on supported bilayers, as demonstrated by total internal reflection fluorescence microscopy (TIRFM) and rapid fluorescence recovery after photobleaching (FRAP, exponential recovery time constant τ = 1.3 s) (*Figure 1B*, and *Figure 1—figure supplement 1A*). Addition of 1 µM Nck causes no change in the distribution of p-Nephrin on the membrane, despite clear association of Nck with the bilayer (*Figure 1—figure supplement 1B*). Similarly, 1 µM of an N-WASP construct containing the basic proline-rich and VCA regions of the protein (residues 183–193, 273–501, N-WASP hereafter) does not change the p-Nephrin distribution. However, addition of 1 µM Nck and 1 µM N-WASP together causes p-Nephrin to organize into micron-sized clusters (*Figure 1B*, *Video 1*). Unphosphorylated Nephrin remains uniformly distributed under these conditions (not shown), indicating that clustering requires binding the Nck SH2 domain to pTyr sites on Nephrin. Labeling of Nck or N-WASP with fluorophores (Alexa 568 or Alexa 647, respectively) shows that the clusters contain all three protein components (*Figure 1—figure supplement 1C*). Quantitative analysis indicated that the clustered regions contain up to fourfold higher density of p-Nephrin than the surrounding regions of the bilayer (*Figure 1C*). Note that much higher concentrations of Nck and N-WASP (~40 µM and ~15 µM, respectively [*Li et al., 2012*]) are required to form phase-separated droplets in solution than to induce p-Nephrin clustering on membranes. Thus, clustering does not involve adhesion of pre-existing three dimensional Nck/N-WASP droplets to membrane-bound p-Nephrin, but rather de novo assembly of the proteins together on the bilayer surface. Further, the DOPC/DOGS lipids in our experiments do not phase-separate, indicating that clustering is independent of lipid phase separation.

To understand the concentration dependence of cluster formation, we fixed the concentration of N-WASP at 500 nM and the density of p-Nephrin at 2700 ± 200 molecules/µm$^2$ (see density control and measurement in 'Materials and methods', also *Figure 2—figure supplement 1, 2*) and added increasing concentrations of Nck. We used two measures to determine the onset of clustering. First, we used a thresholding approach

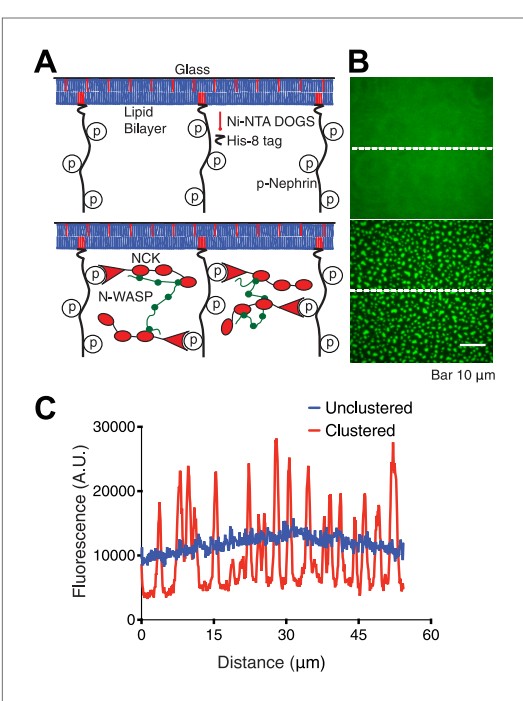

**Figure 1**. Reconstitution of p-Nephrin clusters on supported lipid bilayers. (**A**) Cartoon illustrating the interaction of triply-phosphorylated His$_8$-tagged Nephrin (p-Nephrin) with its partners Nck and N-WASP. Top panel illustrates p-Nephrin attached to bilayers. Bottom panel illustrates the model for clustered p-Nephrin, upon Nck and N-WASP binding. (**B**) Top: TIRF image of Alexa488-labeled p-Nephrin attached to a supported DOPC lipid bilayer doped with 1% nickel-chelating lipid (Ni$^{2+}$-NTA DOGS), (corresponding to panel **A**, top). Bottom: TIRF image of analogous membrane-attached Alexa488-labeled p-Nephrin after addition of 1 µM Nck and 1 µM N-WASP (corresponding to panel **A**, bottom). (**C**) Line-scans of the images in panel **B**, at the positions depicted by the white dotted lines.

The following figure supplement is available for figure 1:

**Figure supplement 1**. p-Nephrin, Nck and N-WASP colocalize to clusters formed on fluid supported lipid bilayers.

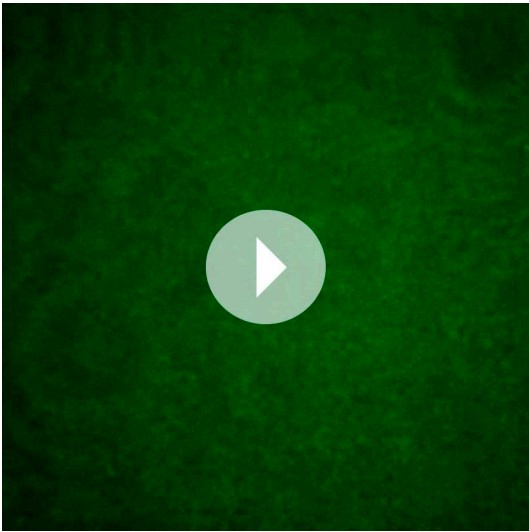

**Video 1**. Addition of Nck and N-WASP to p-Nephrin produces macroscopic clusters on supported bilayers. Time-lapse images taken immediately after adding 1 µM Nck and 1 µM N-WASP to p-Nephrin Alexa488. Images were captured every minute.

to identify and quantify bright regions of the membrane, which we define as clusters. As detailed in 'Materials and methods', two different thresholding procedures gave virtually identical results in this approach. After thresholding, we calculated the fraction of total membrane fluorescence intensity that is present in the clusters. As a second independent approach, we determined the variance of the fluorescence signal across the bilayer image, which also increases as bright regions form. Using either approach, we found that p-Nephrin clusters appear in a highly non-linear fashion as Nck concentration in solution increases. Clusters are essentially absent at low concentrations of Nck but form quite sharply once a critical concentration is reached (~200 nM, *Figure 2A*). We note that the sharp increase in variance and the coincidence of the critical concentration measured by both methods speak against the possibility that small clusters are forming in a more gradual fashion but are too dim to be recognized by the thresholding approach. The average density of p-Nephrin on the membrane does not change during the titration (*Figure 2—figure supplement 2*). We define the concentration at which fractional intensity and variance begin increasing as the clustering concentration. The highly cooperative nature of the cluster formation on bilayers is reminiscent of the sharp phase transitions observed in forming p-Nephrin/Nck/N-WASP liquid droplets in three-dimensional solutions (*Li et al., 2012*). The clusters are distributed randomly (Gaussian distribution) across the membrane (*Figure 2B*), consistent with a stochastic assembly process, where the clusters are nucleated and grow independent of one other (*Dill and Bromberg, 2003*). The clusters also show a broad range of sizes that can be fit well to an exponential distribution (*Figure 2C*), similar to that observed for stochastically assembled chemotaxis receptors in bacteria (*Greenfield et al., 2009*). These properties suggest a stochastic process of cluster formation in our system. In contrast, clusters of GPI-anchored proteins in cells do not show a Gaussian spatial distribution nor a broad size distribution, indicating their active control by the cortical actin cytoskeleton (*Goswami et al., 2008*).

When experiments are performed at ~fivefold higher initial density of p-Nephrin on the membrane, the morphology of the clusters changes significantly. Distinct puncta are no longer observed, and the clustered regions span the entire field of view (*Figure 2D*). These data are consistent with low- and high-density p-Nephrin phase separating via nucleation and spinodal decomposition mechanisms, respectively (*Dill and Bromberg, 2003*), as observed in non-biological phase separating systems in material science (*Zinke-Allmang et al., 1992*). Together, these data strongly suggest that the clustering of p-Nephrin occurs through a phase transition of the molecules on the surface of the membrane in response to binding of Nck and N-WASP.

We next examined the dynamic behaviors of the p-Nephrin clusters. Individual clusters are irregularly shaped, indicating that they possess low line tension. On short timescales, the edges of clusters show substantial fluctuations, extending and retracting in seconds (*Video 2*). On timescales of minutes, these fluctuations lead to coalescence of small clusters into increasingly larger structures (*Figure 3A*, *Videos 1, 2*). We also rarely observe apparent fission events, where a larger cluster seems to split into two smaller structures (*Video 2*). These behaviors suggest that p-Nephrin clusters are fluid-like. The size distribution of the clusters depends on the initial p-Nephrin density and time after Nck/N-WASP addition, reflecting variable contributions of nucleation, growth through monomer addition and coalescence, and Ostwald ripening throughout the process (*Zinke-Allmang et al., 1992*). At lower density (2500 molecules/µm$^2$) the distribution is exponential at all times we examined (*Figure 3—figure supplement 1*), while at higher density (4000 molecules/µm$^2$) the distribution

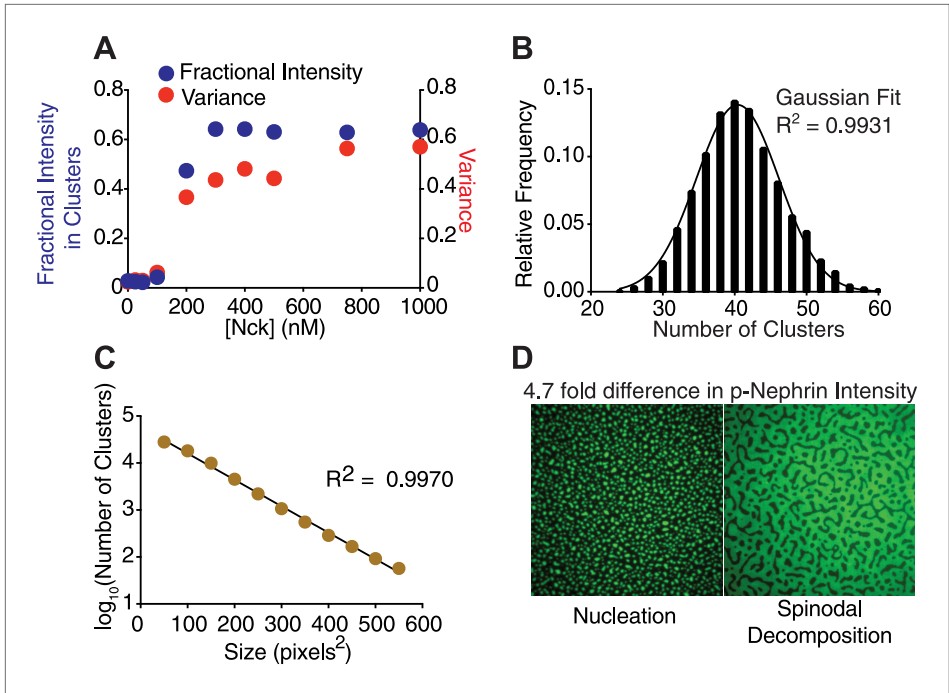

**Figure 2**. Nephrin clusters are created via a two-dimensional phase-transition. (**A**) Fractional intensity in clusters (blue symbols, left ordinate) and signal variance (red symbols, right ordinate) of p-Nephrin fluorescence on a DOPC bilayer as a function of Nck concentration for 500 nM N-WASP and total p-Nephrin density of ~2700 molecules/μm². (**B**) Relative frequency with which a given number of clusters are found within 93 randomly selected 56 × 56 μm regions of a bilayer formed using ~2500 molecules/μm² Alexa488-labeled p-Nephrin, 1 μM Nck, and 1 μM N-WASP. (**C**) Size distribution of clusters formed using ~2500 molecules/μm² Alexa488-labeled p-Nephrin, 1 μM Nck, and 1 μM N-WASP. (**D**) Puncta formed using 1 μM Nck, 1 μM N-WASP, and low (left) or 4.7-fold higher (right) density of p-Nephrin. Images were autocontrasted for clarity.

The following figure supplements are available for figure 2:

**Figure supplement 1**. Quantitative analysis of the measurement and control of His₈-p-Nephrin density on supported lipid bilayers.

**Figure supplement 2**. Quantification of average p-Nephrin density on the bilayer for every titration point shown in *Figure 2A*.

follows a power law (*Figure 3—figure supplement 2*). At a given time after Nck/N-WASP addition, higher density produces a larger average cluster size and correspondingly a larger fraction of total area covered by the clusters (*Figure 3—figure supplement 3A,B*, respectively), most likely due to the larger degree of coalescence at higher cluster densities. A detailed mechanistic understanding of these behaviors will be goal of future efforts.

We next used fluorescence recovery after photobleaching (FRAP) to examine the dynamics of the three proteins that compose the clusters. In individual experiments, we labeled either p-Nephrin, Nck, or N-WASP with Alexa488 and examined FRAP behavior of the labeled component. Within the clusters, each of the proteins recovers nearly fully in tens to hundreds of seconds (*Figure 3B*). Thus, even though the clusters themselves are persistent for hours, the individual components exchange with the surroundings on time-scales of seconds to minutes. The recovery profiles can all be fit to a double-exponential but do not fit to a single-exponential (see 'Materials and methods' for F-test statistics). N-WASP shows recovery time constants of τ-fast = 2.6 s (37%) and τ-slow = 43 s (63%). Nck recovers with τ-fast = 1.6 s (49%) and τ-slow = 72 s (51%). p-Nephrin recovers with τ-fast = 86 s (76%) and τ-slow = 526 s (24%). In the non-clustered regions, p-nephrin recovery can be fit well to a single-exponential, with τ = 31 s, similar to the fast phase in the clusters but appreciably slower than recovery in the absence of Nck/N-WASP, where τ = 1.3 s (*Figure 1—figure supplement 1A*). In independent

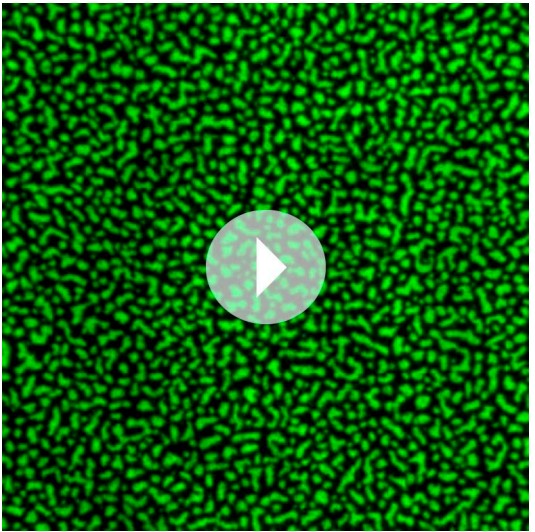

**Video 2**. Clusters are dynamic. Time-lapse of clusters made from 1 μM Nck and 1 μM N-WASP with p-Nephrin Alexa488 on the membrane. Images were captured every 30 s. In addition to fusion events, the clusters also occasionally undergo fission.

experiments we found that the dissociation of p-nephrin from the membrane occurs much more slowly than these rates (τ = 2080 s, *Figure 2—figure supplement 1D*), indicating that the FRAP recovery of the protein is largely due to two-dimensional diffusion within the bilayer. By contrast, Nck and N-WASP likely recover through a combination of diffusion in the plane of the bilayer as well as binding and dissociation from the membrane. We recognize that the kinetic processes here must represent the convolution of multiple molecular processes, given the complex oligomeric/polymeric nature of the clusters (see below). Nevertheless, the data suggest that both the clustered regions and non-clustered regions contain small assemblies that slow p-nephrin dynamics relative to its free diffusion in the bilayer. The clustered region likely contains additional assemblies that are larger and have greater degrees of crosslinking that appreciably slow dynamics further.

Together, our data show that upon recruitment of Nck and N-WASP, membrane-bound p-nephrin undergoes a sharp thermodynamically controlled phase transition to produce dense dynamic puncta on the membrane.

## Phase separation occurs through polymerization of p-nephrin, Nck and N-WASP

Our previous data suggested that three dimensional phase separation in the p-nephrin/Nck/N-WASP system occurred concomitantly with a sol–gel transition, producing macroscopic non-covalent polymers within the liquid phase boundary. Evidence for polymerization came in part from studies of the dependence of critical concentration and dynamics on the valencies and affinities of the interacting species. To examine whether such polymerization is also occurring in the two-dimensional system, we initially compared the critical concentrations of singly-, doubly-, and triply-phosphorylated nephrin (Nephrin1pY, Nephrin2pY, and p-Nephrin, respectively; see 'Materials and methods' for specific phosphorylation sites). Previous studies showed that the three nephrin pTyr sites have essentially identical affinities for the Nck SH2 domain (*Blasutig et al., 2008*). Thus, these constructs differ largely in pTyr valency, rather than inherent affinity for Nck. At a membrane density of 1000 molecules/μm$^2$ and in the presence of 500 nM N-WASP, p-Nephrin begins to show clusters at 200–300 nM Nck. Under the same conditions, Nephrin2pY and Nephrin1pY do not cluster even at Nck concentrations greater than 10 μM (*Figure 4A*) nor with their own densities increased to 3000 molecules/μm$^2$. If the concentrations of N-WASP and Nck are increased to 2 μM and 5 μM, respectively, Nephrin2pY produces clusters (*Figure 4—figure supplement 1*). However, even at 5 μM N-WASP and 10 μM Nck, Nephrin1pY does not cluster (*Figure 4—figure supplement 1*). Thus, the valency of nephrin phosphorylation can control the critical concentration for puncta formation, as in the three-dimensional phase separation of this system (*Li et al., 2012*).

We also performed analogous studies of the SH3 valency of Nck. Since the different SH3 domains of Nck have different affinities for the individual PRM sites in N-WASP (Qiong Wu, unpublished observations), we generated a series of Nck analogs, containing one, two, or three repeats of the second SH3 domain of the protein plus the natural SH2 domain [(SH3)$_1$, (SH3)$_2$, and (SH3)$_3$, respectively]. The SH3 domains were separated by the natural linker between the first and second SH3 domains. At 500 nM N-WASP, the trivalent molecule (SH3)$_3$ induces clustering at 200 nM (SH3 module concentration), whereas the di-valent (SH3)$_2$ and monovalent (SH3)$_1$ molecules do not cluster even at concentrations above 10 μM of the SH3 module concentrations (*Figure 4B*). Increasing N-WASP concentration to 5 μM and (SH3)$_2$ concentration to 5 μM (SH3 module concentration) produced clusters, whereas clusters were absent even with 5 μM N-WASP and 5 μM (SH3)$_1$

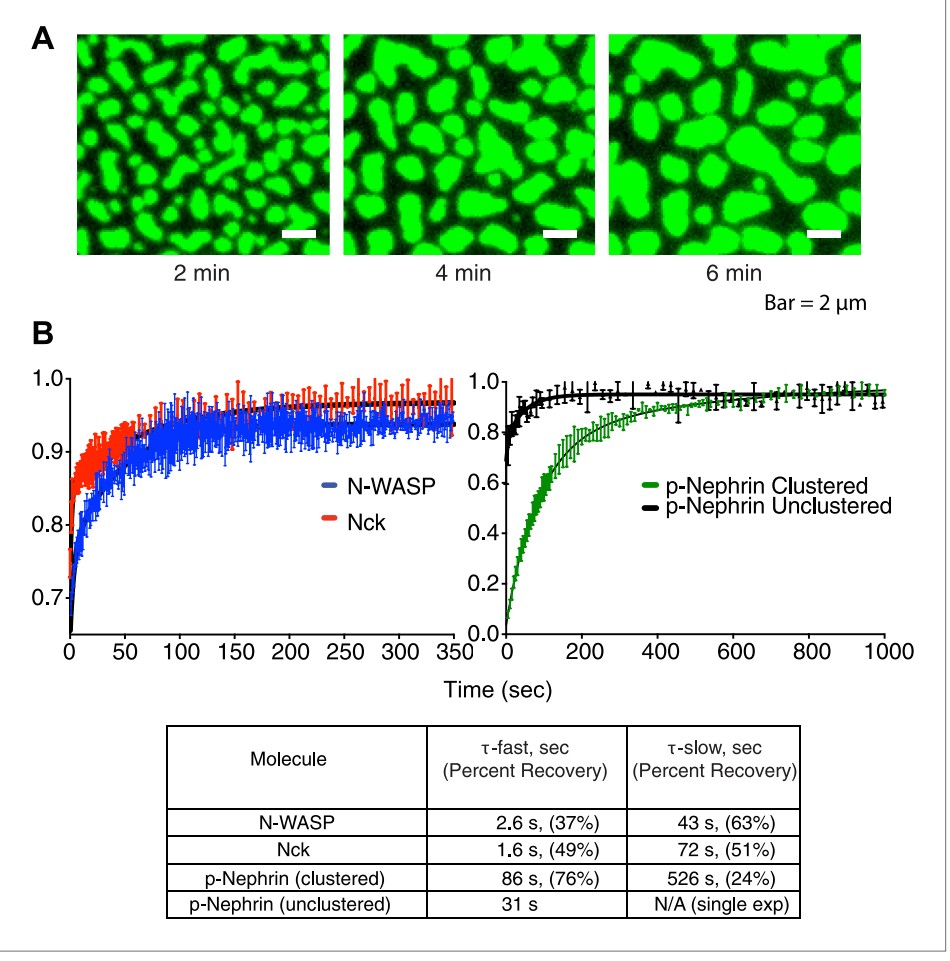

| Molecule | τ-fast, sec (Percent Recovery) | τ-slow, sec (Percent Recovery) |
|---|---|---|
| N-WASP | 2.6 s, (37%) | 43 s, (63%) |
| Nck | 1.6 s, (49%) | 72 s, (51%) |
| p-Nephrin (clustered) | 86 s, (76%) | 526 s, (24%) |
| p-Nephrin (unclustered) | 31 s | N/A (single exp) |

**Figure 3**. Clusters are dynamic. (**A**) Time-lapse TIRF imaging of bilayers containing ~3100 molecules/μm² Alexa488-labeled p-Nephrin after addition of 1 μM Nck and 1 μM N-WASP. Images represent time intervals of 2 min and show coalescence of clusters into larger structures. (**B**) Fluorescence recovery after photobleaching of Nck and N-WASP in clustered regions (left panel, red and blue, respectively) and p-Nephrin in clustered and unclustered regions (right panel, green and black, respectively). FRAP experiments were performed in separate experiments using Alexa-488 labeled p-Nephrin, Nck, or N-WASP. Lines show bi-exponential fits of the data, except for unclustered p-Nephrin, which was fit using a single-exponential. Bars represent standard deviation from three FRAP experiments on a single bilayer. Bottom table lists the parameters obtained from the fitting.

The following figure supplements are available for figure 3:

**Figure supplement 1**. Cluster size-distribution analyses at different times suggest exponential behavior at lower densities.

**Figure supplement 2**. Cluster size-distribution analyses suggest power law behavior at higher densities.

**Figure supplement 3**. Average cluster size is dependent on molecular density.

(*Figure 4—figure supplement 1*). These data demonstrate the strong dependence of clustering on valency of the interacting species.

To determine the effect of SH2–pTyr affinity on the clustering concentrations, we replaced the three pTyr motifs of Nephrin with three repeats of the pTyr motif of the bacterial protein TIR (p-TIR) (*Campellone et al., 2002*). The binding affinity of the p-Nephrin motif to the SH2 domain of Nck is 370 nM, as determined by isothermal titration calorimetry (*Figure 5—figure supplement 1*). For the p-TIR motif the affinity to the SH2 domain is 40 nM. In the presence of p-TIR, at a density of

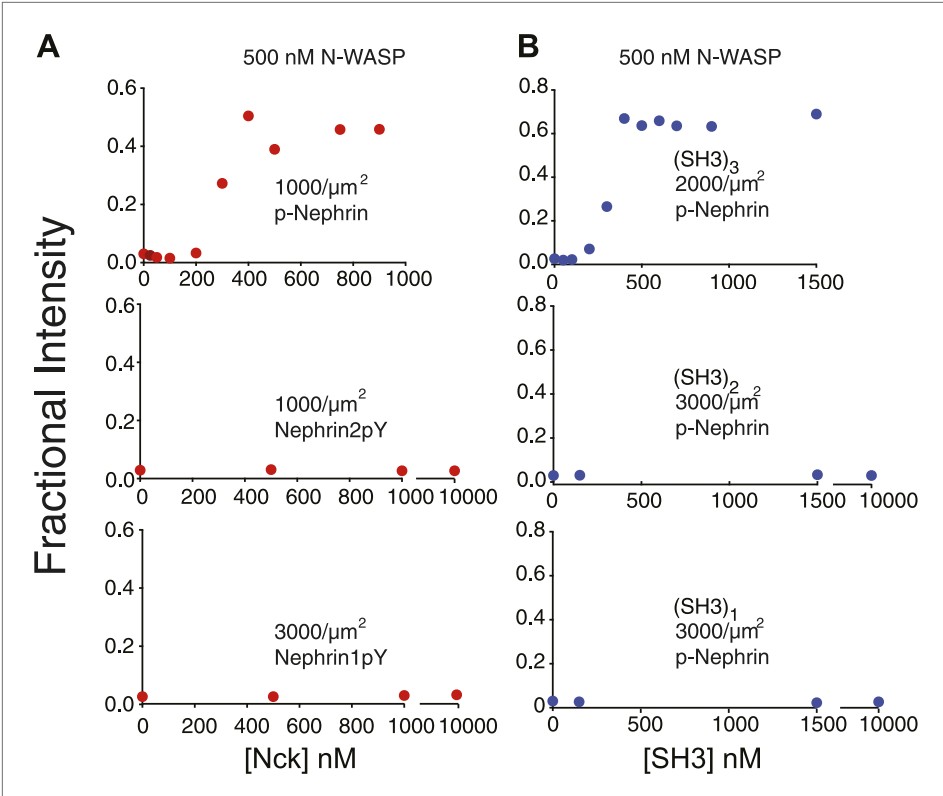

**Figure 4**. Clustering is dependent upon the valency of the interacting motifs. Plots show fractional intensity of fluorescent Nephrin proteins in clusters as a function of Nck protein concentrations for 500 nM N-WASP. (**A**) Top, middle, and bottom panels show data for p-Nephrin 3pY, 2pY, and 1pY, respectively. For these concentrations of N-WASP and Nck, only Nephrin 3pY shows clustering. At 2 µM N-WASP, Nephrin 2pY also clusters when Nck is added (***Figure 4—figure supplement 1***). (**B**) Top, middle, and bottom panels show data for p-Nephrin plus engineered Nck proteins containing 3, 2, or 1 repeat of the second SH3 domain of Nck. For these concentrations/densities of N-WASP/p-Nephrin, only the (SH3)$_3$ protein can induce clustering. At 5 µM N-WASP, (SH3)$_3$ also induces clustering (***Figure 4—figure supplement 1***). Note that the x-axis is Nck concentration in panel **A** but total SH3 domain concentration in panel **B**.

The following figure supplement is available for figure 4:

**Figure supplement 1**. Di-valent molecules are stronger clustering agents than mono-valent molecules.

2000 molecules/µm², the clustering concentration of the trivalent SH3 protein, (SH3)$_3$, is 100 nM, as opposed to 200 nM for p-Nephrin (***Figure 5A***). The higher affinity interaction also slows the recovery of Nck, as FRAP data demonstrate (***Figure 5B***). Fitting to a double-exponential, Nck shows recovery rate constants of τ-fast = 6.5 s (46%) and τ-slow = 89.5 s (54%) when the clusters of p-TIR/Nck/N-WASP were photobleached. However, Nck shows recovery rates of τ-fast = 1.6 s (49%) and τ-slow = 73.2 s (51%) when the clusters of p-Nephrin/Nck/N-WASP were photobleached. The data would be consistent with τ-fast being governed by processes based on dissociation of Nck from pTyr sites on Nephrin/TIR (which are likely slower in the high affinity system) and τ-slow being governed by diffusion of large assemblies in the membrane (which are expected to be similar in the two cases). Together, the data show that both the clustering concentrations and the dynamics of the clusters can be affected by molecular affinities, as expected of a crosslinked polymer network.

Additionally when a higher-affinity monovalent pTyr peptide is added in solution, the clusters dissipate. In the presence of clusters made from 1 µM (SH3)$_3$, 500 nM N-WASP and p-nephrin, we added singly phosphorylated TIR peptide (without a His tag) at 10 µM concentration (***Video 3***, ***Figure 6A***). The clusters disappear within minutes after the addition of the monovalent peptide. The dissolution of the clusters occurs sharply, over a time-span of ~2 min, starting ~7 min after peptide addition (***Figure 6B***).

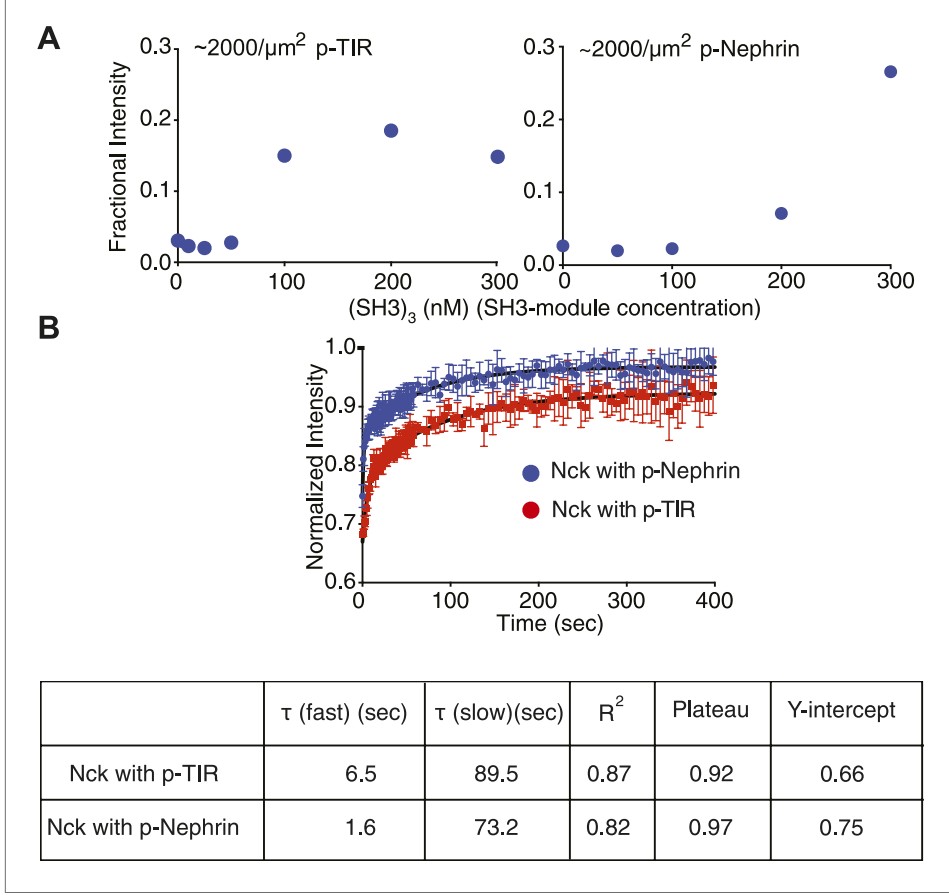

**Figure 5**. Molecular affinities affect macroscopic clustering. (**A**) Fractional intensity of fluorescent pTyr proteins in clusters as a function of SH3 (module) concentrations for 500 nM N-WASP. Left and right panels show data for a p-TIR and p-Nephrin, whose pTyr motifs bind the SH2 domain of Nck with $K_D$ values of 40 nM and 370 nM, respectively. (**B**) Fluorescence recovery after photobleaching (FRAP) for Alexa488-labeled Nck in p-Nephrin clusters (blue) and p-TIR clusters (red). Nck recovers more slowly (larger τ), can be bleached more strongly (Y-intercept) and recovers to a lower value (plateau) with p-TIR than with p-Nephrin, all indicating slower dynamics in clusters with the higher affinity SH2 binding partner. The bars represent standard deviation from three FRAP experiments on a single bilayer.
The following figure supplement is available for figure 5:

**Figure supplement 1**. Measurement of the affinity of Nck for p-TIR and p-Nephrin.

When TIR is titrated from 100 nM to 100 µM, the fractional intensity of the clusters also decreases sharply above 10 µM (*Figure 6C*). These data suggest that the disassembly of the clusters (similar to the formation) is also cooperative.

The favorability of higher valency and higher affinity on clustering, as well as the disruption of the clusters by a mono-valent molecule, suggest that as in the three dimensional droplets the two dimensional clusters form through polymerization (a sol–gel transition) of p-Nephrin, Nck and N-WASP.

## p-Nephrin/Nck/N-WASP clusters promote Arp2/3 complex-dependent actin assembly

We next asked whether the p-Nephrin/Nck/N-WASP clusters can direct actin assembly by the Arp2/3 complex at membranes. We added monomeric actin (10% rhodamine labeled) to the solution above preformed clusters in the presence or absence of the Arp2/3 complex, under conditions that favor actin polymerization. Immediately after addition a small amount of actin, likely monomers, is recruited to the clusters in a relatively uniform fashion (*Figure 7—figure supplement 1A*). After a lag of ~6–15 min

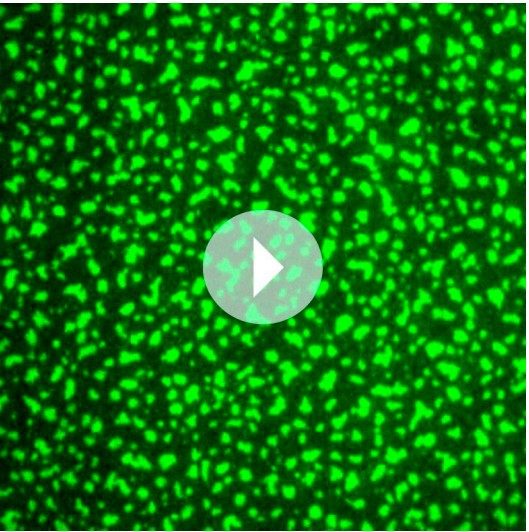

**Video 3**. Mono-valent peptide dissolves clusters. Addition of 10 μM 1pY—TIR causes the clusters of 1 μM (SH3)$_3$ and 500 nM N-WASP to dissipate. Images were captured every 30 s.

(see below), actin filaments then assemble on the clusters over a time course of approximately 100 min, as visualized by phalloidin 647 staining (*Figure 7—figure supplement 1A*).

In the absence of the Arp2/3 complex, actin filaments are formed only sparsely in the field of view (*Figure 7—figure supplement 1B*). In the presence of the Arp2/3 complex, actin appears on the clusters much more rapidly and to a much greater degree (*Figure 7A*, *Figure 7—figure supplement 1B*). The lag time between the initial weak recruitment of actin and the appearance of robust actin fluorescence (presumably representing filaments) varies substantially between clusters (*Figure 7A,B*). Some clusters show increased actin after only 6 min, while others remain devoid of additional actin until 10–15 min into the reaction. This behavior appears to be stochastic, and the lag time does not show any obvious correlation with size or density of the Nephrin clusters (*Figure 7B*). Regardless of when filament assembly begins on a cluster, once it does begin, actin intensity rapidly increases, typically reaching a plateau in less than 10 min (*Figure 7C*). This behavior likely reflects strong positive feedback due to activation of the Arp2/3 complex by actin filaments (*Machesky et al., 1999*).

As the reaction proceeds, the morphology of the Nephrin clusters changes dramatically, without appreciable changes on overall intensity (except the slow decrease due to photobleaching). Actin fluorescence remains coincident with Nephrin throughout this process, indicating that the signaling molecules are reorganized by the assembling filaments. Shortly after the appearance of actin on a cluster, the structure changes from having relatively rounded edges to having many thin hair-like projections from its periphery. These projections coalesce over time to give the puncta star-like morphologies. For reasons we cannot currently explain, between 42 and 45 min, well after all of the clusters have recruited significant actin, there is a dramatic change in cluster morphology, such that the puncta appear to shatter into a large number of short linear structures (*Figure 7—figure supplement 2*). This change in Nephrin morphology coincides with a sharp increase in the total actin localized to the TIRF field/membrane but no change in total Nephrin fluorescence.

These data demonstrate the p-Nephrin/Nck/N-WASP clusters can effectively promote actin filament assembly through the Arp2/3 complex (which is presumably recruited to the membrane through binding N-WASP). As the filaments assemble, they cause substantial changes in the morphology of the clusters. This feedback between actin and the signaling proteins that promote its assembly can control the micron-scale morphology of the entire pathway.

## Discussion

### Polymerization and concomitant phase separation as a general mechanism to create membrane clusters

We have shown here that membrane-bound phosphorylated Nephrin can form micron-size clusters through interactions with Nck and N-WASP. The clusters form through a thermodynamic phase transition that is driven by oligomerization/polymerization of the proteins through their modular binding domains. The occurrence of a phase transition is supported by the sharpness with which clusters appear as Nck concentration is increased and the temporal sharpness of cluster disappearance after a monovalent competitor is added. The importance of polymerization/oligomerization is supported by the valency- and affinity-dependence of the critical concentration and also by the dissolution of clusters by a monovalent competitor. The clusters appear to be polymers/oligomers of the three proteins, as evidenced by the dependence of FRAP rate on the affinity of the Nck SH2 domain for the pTyr sites on

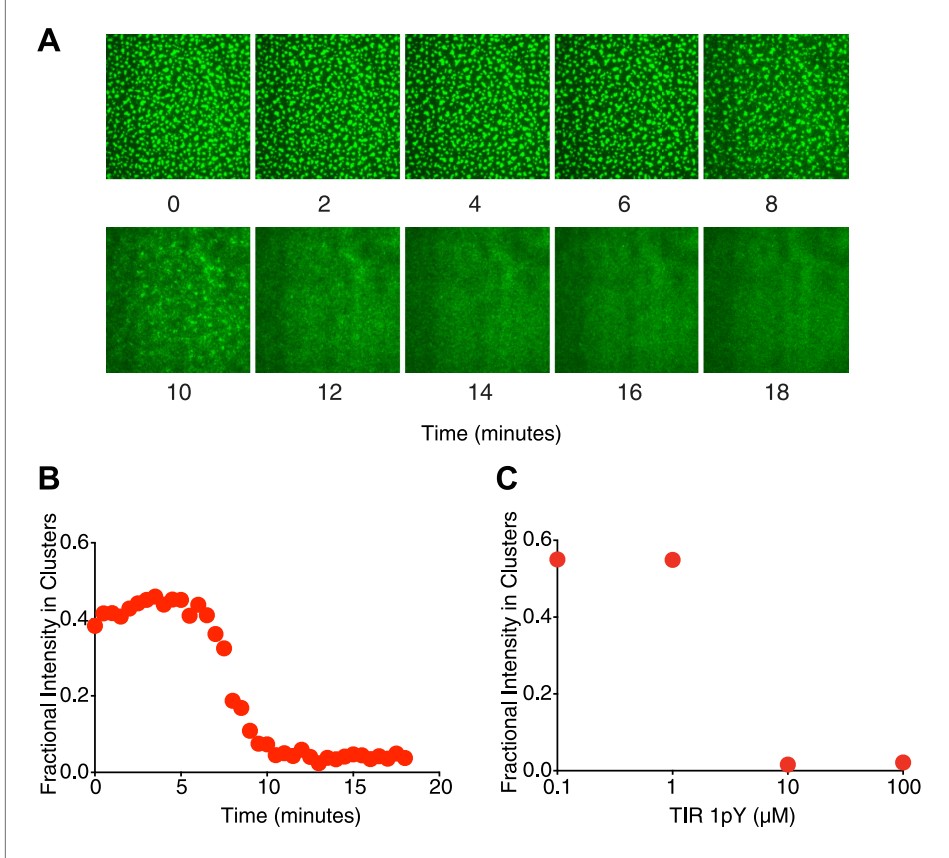

**Figure 6**. Mono-valent pTyr peptide can eliminate clusters. (**A**) Time course following addition of 10 μM of a monovalent pTyr peptide derived from TIR (with $K_D$ of 40 nM for the Nck SH2 domain) to clusters formed from p-Nephrin /(SH3)$_3$/N-WASP. (**B**) Time course of the fractional p-Nephrin intensity in clusters after addition of the TIR peptide. (**C**) Equilibrium fractional intensity of the p-Nephrin clusters as a function of p-TIR peptide concentration, performed in the presence of 1 μM (SH3)$_3$ and 500 nM N-WASP.

Nephrin. The clusters assemble actin through the Arp2/3 complex and can themselves be dynamically remodeled by the resultant filament network. Our work demonstrates that, as in three-dimensional systems, multivalent polymerization and phase separation can control micron-scale spatial organization (and likely biochemical activity) of signaling pathways.

This process may contribute generally to the organization of signaling receptors. The cytoplasmic tails of many receptors are rapidly phosphorylated on multiple tyrosine residues upon stimulation by extracellular ligands (*Roche et al., 1996*; *Hunter, 2000*; *Palmer et al., 2002*; *Schlessinger, 2000*; *Houtman et al., 2006*; *Kaushansky et al., 2008*; *Wagner et al., 2013*). This often occurs concomitant with concentration of the receptors into micron-sized puncta (*Douglass and Vale, 2005*; *Salaita et al., 2010*). Where examined, these puncta persist over many minutes, but exchange molecules in seconds with the surroundings, similar to the p-Nephrin/Nck/N-WASP puncta we have generated here. Further, many of these receptors have been shown, through combinations of biochemistry and genetics, to use the pTyr modifications to engage signaling networks composed of proteins with multiple modular binding domains, often (but not exclusively) combinations of SH2 domains, SH3 domains, PRMs, and additional pTyr sites. Examples of processes controlled by such pathways include T cell activation (*Lee et al., 2003*; *Dustin et al., 2010*), invadopodia formation (*Oser et al., 2010*; *Bergman et al., 2014*), myoblast fusion (*Abmayr and Pavlath, 2012*), neurite self-avoidance (*Chen and Maniatis, 2013*), and cell–matrix interactions through focal adhesions (*Hoffmann et al., 2014*). The molecules that control these processes have the capacity to function analogous to the Nephrin/Nck/N-WASP system studied here. We hypothesize that coupled polymerization and phase separation may contribute to the formation of macroscopic puncta in these systems and others that are composed similarly.

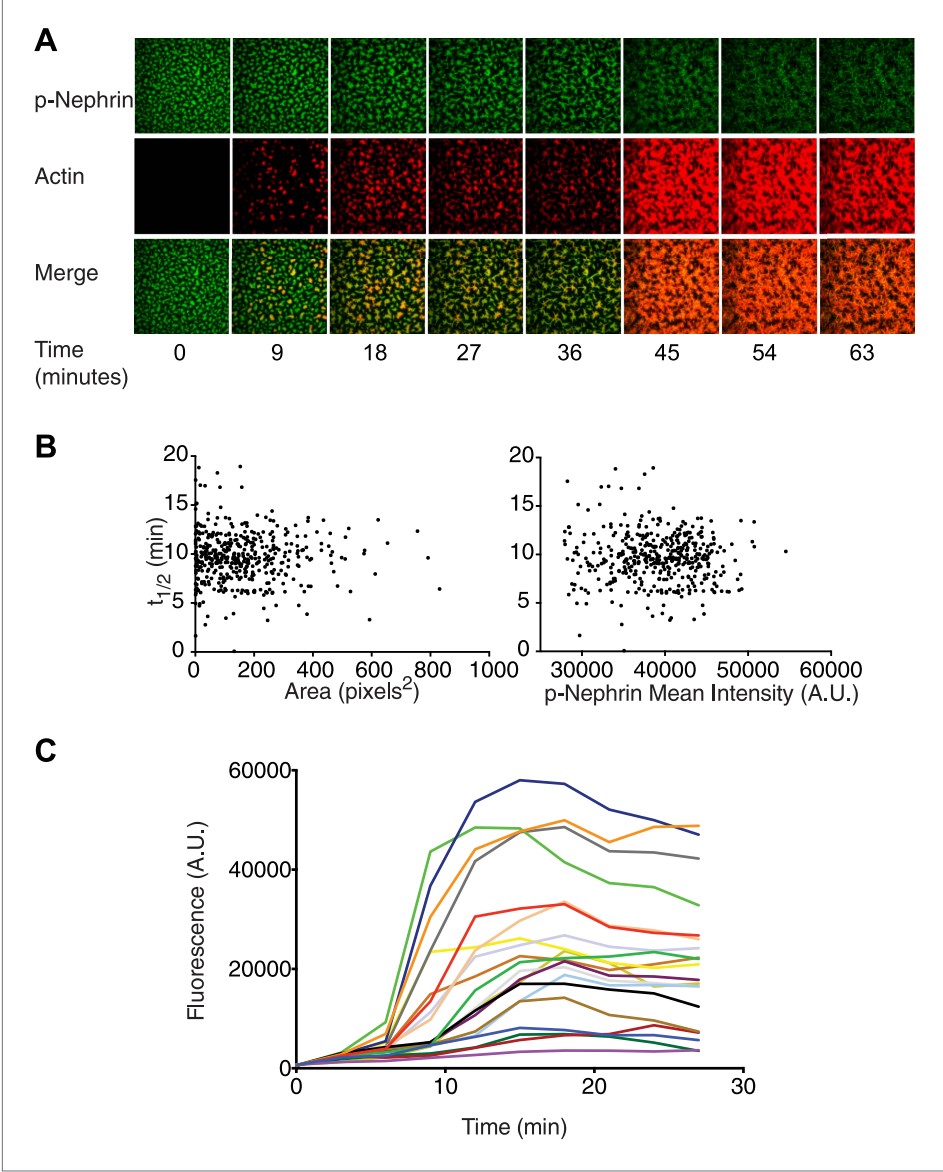

**Figure 7**. Actin assembles specifically on p-Nephrin/Nck/N-WASP clusters. (**A**) Alexa488-labeled p-Nephrin (2200 molecules/μm²) was clustered by addition of 2 μM N-WASP and 1 μM Nck. Images show time course of p-Nephrin (top row), actin (middle row) and merge (bottom row) after addition of 10 nM Arp2/3 complex and 1 μM actin (10% rhodamine labeled). (**B**) Half-times of actin assembly as a function of surface area (left-panel) and p-Nephrin intensity (right-panel) in individual clusters. Half-times were calculated using the data for the first 27 min of the time-lapse. (**C**) Fluorescence of rhodamine-actin on individual clusters as a function of time for 20 representative clusters. Individual curves represent average intensity across an individual cluster.

The following figure supplements are available for figure 7:

**Figure supplement 1**. Actin localizes to and assembles on the clusters in an Arp2/3 dependent manner.

**Figure supplement 2**. Actin assembly reorganizes p-Nephrin clusters.

We note that polymerization does not strictly require multivalency at the level of an individual receptor tail. At high densities, a receptor containing a single motif behaves effectively in multivalent fashion and can then cluster through interactions with multivalent ligands. For example, proteins with multiple PDZ domains interact with voltage-gated Kv1.4 channels, which are found in clusters at the

cell-surface (*Burke et al., 1999*). Further, membrane receptors are often oligomeric in nature. For example, EGF receptors have been reported to form pre-formed oligomers in the absence of ligand (*Clayton et al., 2008*), effectively increasing the valency of their cytoplasmic tails. Thus, there are a variety of ways that the basic concept of multivalent polymerization and phase separation could be manifested in specific signaling systems.

This behavior appears to be particularly prominent in actin regulatory pathways. These often contain the adaptor proteins Nck or Crk/CrkL, linked upstream to pTyr-containing proteins and downstream to proline-rich proteins including members of the WASP family (*Buday et al., 2002*; *Antoku et al., 2008*; *Noy et al., 2012*; *Chaki and Rivera, 2013*; *Kaipa et al., 2013*). In this regard, it is notable that over half of the 29 known ligands of the Nck SH2 domain contain two or more (up to 16 in p130 CAS) predicted and/or demonstrated Nck binding sites (*Lettau et al., 2009*). Further, almost all WASP proteins have large proline-rich regions with multiple SH3-binding PRMs (*Padrick and Rosen, 2010*). The only exception is WASH, which has a small proline-rich region. However, WASH is constitutively associated with the Fam21 protein, which has a large disordered tail that contains 21 so-called LFa peptide motifs that can bind the membrane-associated retromer complex (*Derivery et al., 2009*; *Gomez and Billadeau, 2009*; *Jia et al., 2010, 2012*). Thus WASH may have a conceptually similar but molecularly distinct mechanism of assembling into large structures. This general behavior suggests that clustering may play an important role in spatial and temporal control of actin dynamics. Consistent with this idea, several groups have demonstrated that increased density of WASP proteins corresponds to increased activity towards the Arp2/3 complex. Pantaloni and colleagues have shown that the rate of actin-based motility of N-WASP-coated beads increases non-linearly with increasing WASP density (*Wiesner et al., 2003*). Similarly, Ditlev et al. have shown through modeling and cell-based experiments that actin assembly activity scales with the square of N-WASP density at the plasma membrane (*Ditlev et al., 2012*). Finally, Padrick et al. showed that a natural consequence of the 2:1 stoichiometry of the WASP:Arp2/3 complex during filament nucleation is that actin assembly activity should increase with the size of WASP clusters and thus the density of WASP at membranes (*Padrick et al., 2008*). These observations, together with our data here, suggest that clustering of receptors and their proximal adaptors may provide a mechanism of concentrating WASP proteins into high-density puncta and thus increasing their activation of the Arp2/3 complex, providing local bursts of actin filament assembly.

The mechanism we have described is not exclusive of, and in fact is expected to act cooperatively with, many other mechanisms that have been proposed to explain receptor clustering. Interactions between extracellular domains, as proposed for cadherins and Eph receptors (*Himanen et al., 2007*; *Wu et al., 2011*; *Seiradake et al., 2013*), will be thermodynamically coupled to assembly of intracellular oligomers/polymers. Similarly, interactions of receptor transmembrane regions with specific lipids, which can promote concentration of receptors into nano-domains enriched in those lipids, should also be thermodynamically coupled to the clustering of receptor cytoplasmic tails (*Lingwood and Simons, 2010*). Further, ATP-dependent clustering of the cortical acto-myosin system, which promotes oligomerization of GPI-anchored proteins (*Sharma et al., 2004*; *Goswami et al., 2008*), could also promote assembly of a phase separated multivalent network if any components of that network can bind to the cytoskeleton. In this case, the dynamic rearrangements of the acto-myosin system could also control the properties of the signaling clusters (e.g., cluster size and/or lifetime of clusters) and maintain them away from equilibrium. It is important to note that while weak interactions between extracellular domains or between transmembrane regions and lipids or between receptors and the cytoskeleton may not on their own produce significant oligomerization of receptors, these energies could have substantial effects when combined with energies of clustering. Phrased differently, these other interactions could have strong effects on the critical concentrations (or the degree of receptor phosphorylation) needed for phase separation/clustering through adaptor-based intracellular interactions. For any particular system, or for a single system under different conditions, these various mechanisms are likely to be used to different degrees to promote the organization of receptors into macroscopic structures.

## Functional implications of clustering through multivalent phase separation

The ability of membrane receptors to cluster through multivalent phase separation could have a number of functional implications in cells. The process will generate a sharp switch between different states, which will depend on the concentrations of at least two (and possibly several) species, as well

as the degree of receptor phosphorylation in pTyr-dependent cases. Thus, the switch could be tightly controlled, either through relatively slow changes in protein concentration or more rapidly through changes in receptor phosphorylation or oligomerization by extracellular ligands. The phase-separated state will have different density, composition, and dynamics from the surrounding regions of the membrane, each of which could have functional consequences. In the case of actin regulatory systems, we and others have shown that because WASP proteins bind (and activate) Arp2/3 complex in 2:1 fashion, increasing density of WASP proteins leads to non-linear increases in actin assembly activity (*Padrick et al., 2008*; *Ditlev et al., 2012*). Thus, clustering should provide not only spatial organization of the actin filament network (decreasing spatial noise [*Grecco et al., 2011*]) but also increased biochemical signaling activity. This should be true for any signaling system that requires multiple simultaneous or sequential events to generate downstream outputs. The enhancement due to clustering would be particularly strong for systems with positive feedback, as in Arp2/3 complex–actin pathways. In addition to the polymer components themselves, other proteins and lipids could be concentrated into or excluded from the phase-separated structure. This partitioning could be dictated by both specific interactions (e.g., a monovalent SH3 protein could be recruited to the p-Nephrin/Nck/N-WASP clusters by binding the PRMs of N-WASP) as well as non-specific electrostatic and/or hydrophobic interactions with the polymer matrix. The collection of these molecules would then produce a distinct biochemical environment from the surrounding regions, favoring or disfavoring certain reactions or afford specificity to signaling pathways. Since the clusters are temporally stable but readily exchange molecules with the surroundings, they could potentially act as sites of enzymatic modification and release. Finally, the structural and dynamic features of the polymer matrix could also influence the rates and/or specificities of reactions that occur within the clusters.

### Nephrin oligomerization as a mechanism to organize the slit diaphragm

Recent data have shown that Nephrin is constitutively phosphorylated in the slit diaphragm between podocytes of the kidney (*Jones et al., 2009*; *New et al., 2013*). Previous data showed that the loss of Nck disrupts the filtration capacity of the diaphragm, concomitant with the loss of cortical actin filaments (*Jones et al., 2006*). These observations suggest that the pathway from p-Nephrin to actin, and by inference the polymeric network we have described here, is important in maintaining the slit diaphragm. The extracellular portion of Nephrin is composed of multiple IgG domains and FNIII domains. These have been suggested to self-associate, both in trans across the slit diaphragm and in cis within individual cells (*Gerke et al., 2003*). The latter should promote polymerization and phase separation of the actin pathway components. Thus, this system may be a case where interactions on both sides of the plasma membrane act cooperatively to produce a polymeric structure with both extracellular functions (the filtration barrier) and intracellular functions (signaling to actin).

### Conclusion

In summary, we have shown that interactions between multivalent proteins at membranes can lead to concomitant polymerization and phase separation, generating micron-size clusters. Although only demonstrated here for the p-Nephrin/Nck/N-WASP system, the analogous construction of many signaling pathways suggests that this behavior could be quite general, and relevant to many biological processes. Polymerization and phase separation at membranes could impart spatial organization on these pathways and afford them strongly non-linear activities. Further work in vitro and in vivo will be necessary to determine the extent to which these effects are important in specific biological processes.

## Materials and methods

### Protein expression and purification, phosphorylation of nephrin

Information on different constructs is provided in *Table 1*. Maltose binding protein (MBP)-tagged His$_8$-Nephrin and its mutants were expressed in BL21(DE3)T1R cells at 18°C through overnight induction with 1 mM IPTG. Cells were collected by centrifugation and lysed by cell disruption (Emulsiflex-C5, Avestin, Ottowa, ON, Canada) in 20 mM Tris, pH 8, 20 mM imidazole, 150 mM NaCl, 5 mM βME, 0.01% NP-40, 10% glycerol, 1 mM PMSF, 1 µg/ml antipain, 1 mM benzamidine and 1 µg/ml leupeptin. The cleared lysate was applied to Ni-NTA agarose (Qiagen, Venlo, Netherlands), washed with the lysis buffer containing 300 mM NaCl and 50 mM imidazole, and eluted with the same buffer but containing 150 mM NaCl and 300 mM imidazole. The MBP was removed with TEV protease treatment at 4°C for 16 hr or at room-temperature for 2 hr. The protein was further purified using a Source 15Q column

**Table 1.** Information on the protein constructs used in this study

| Proteins | Sequence information | Notes |
|---|---|---|
| Nck | GHMAEEVVVAKFDYVAQQEQELD IKKNERLWLLDDSKSWWRVRNSMNK TGFVPSNYVERKNSARKASIVKNLK DTLGIGKVKRKPSVPDSASPADDSF VDPGERLYDLNMPAYVKFNYMAERED ELSLIKGTKVIVMEKCSDGWWRGSYN GQVGWFPSNYVTEEGDSPLGDHVGSL SEKLAAVNNLNTGQVLHVVQALYPFS SSNDEELNFEKGDVMDVIEKPENDPEW WKCRKINGMVGLVPKNYVTVMQNNPLT SGLEPSPPQCDYIRPSLTGKFAGNPWY YGKVTRHQAEMALNERGHEGDFLIRDS ESSPNDFSVSLKAQGKNKHFKVQLKET VYCIGQRKFSTMEELVEHYKKAPIFTS EQGEKLYLVKHLS | Human, WT, residues 1–377 |
| N-WASP BPVCA | GSEFKEKKKGKAKKKRAPPPPPPSRGG PPPPPPPPHSSGPPPPPARGRGAPPPP PSRAPTAAPPPPPPSRPGVVVPPPPPNR MYPHPPPALPSSAPSGPPPPPPLSMAGS TAPPPPPPPPPPPGPPPPPGLPSDGDHQ VPASSGNKAALLDQIREGAQLKKVEQNS RPVSCSGRDALLDQIRQGIQLKSVSDGQE STPPTPAPTSGIVGALMEVMQKRSKAIHS SDEDEDDDDEEDFEDDDEWED | Rat, residues 183–193 fused to 273–501 |
| Nck (cysteine-modified) | GHMCMAEEVVVAKFDYVAQQEQELDIKK NERLWLLDDSKSWWRVRNSMNKTGFVPSNY VERKNSARKASIVKNLKDTLGIGKVKRKPS VPDSASPADDSFVDPGERLYDLNMPAYVKF NYMAEREDELSLIKGTKVIVMEKSSDGWWR GSYNGQVGWFPSNYVTEEGDSPLGDHVGSL SEKLAAVNNLNTGQVLHVVQALYPFSSSND EELNFEKGDVMDVIEKPENDPEWWKARKING MVGLVPKNYVTVMQNNPLTSGLEPSPPQSDY IRPSLTGKFAGNPWYYGKVTRHQAEMALNER GHEGDFLIRDSESSPNDFSVSLKAQGKNKHF KVQLKETVYSIGQRKFSTMEELVEHYKKAPIF TSEQGEKLYLVKHLS | Human, residues 1–377, with mutations: C139S, C232A, C266S, C340S |
| Nephrin3Y | GGSLEHHHHHHHGGSCGGSGGSGGSGG SHLYDEVERTFPPSGAWGPLYDEVQMGPW DLHWPEDTFQDPRGIYDQVAGD | Human, residues 1174–1223, with mutations: Y1183F, Y1210F |
| Nephrin2Y | GGSLEHHHHHHHGGSCGGSGGSGGSGGS HLFDEVERTFPPSGAWGPLYDEVQMGPWD LHWPEDTFQDPRGIYDQVAGD | Human, residues 1174–1223, with mutations: Y1176F, Y1183F, Y1210F |
| Nephrin1Y | GGSLEHHHHHHHGGSCGGSGGSGGSGGSHL FDEVERTFPPSGAWGPLYDEVQMGPWDLHWP EDTFQDPRGIFDQVAGD | Human, residues 1174–1223, with mutations: Y1176F, Y1183F, Y1210F, Y1217F |
| TIR3Y | GGSLEHHHHHHHGGSCGGSGGSGGSGGSHM HIYDEVAADPPPSGAWGHIYDEVAADPWD LHWPEDTFQDPRHIYDEVAADP | Human Nephrin, with pTyr sites replaced by those in EPEC Tir protein (underlined) |
| (SH3)₃ | GH**MPAYVKFNYMAEREDELSLIKGTKVIVME KSSDGWWRGSYNGQVGWFPSNYVTEEGDSPL**SARKASIVKNLKDTLGIGKVKRKPSVPDSA SPADDSFVDPGERLYDLN**MPAYVKFNYM AEREDELSLIKGTKVIVMEKSSDGWWRGSYNGQV GWFPSNYVTEEGDSPL**SARKASIVKNLK DTLGIGKVKRKPSVPDSASPADDSFVDPGERLY DLN**MPAYVKFNYMAEREDELSLIKGTKV IVMEKSSDGWWRGSYNGQVGWFPSNYVTEEG DSPL**NNPLTSGLEPSPPQCDYIRPSLTGKFAG NPWYYGKVTRHQAEMALNERGHEGDFLIRDS ESSPNDFSVSLKAQGKNKHFKVQLKETVYCI GQRKFSTMEELVEHYKKAPIFTSEQG EKLYLVKHLS | Human, three repeats of the second Nck SH3 domain, plus the Nck SH2 domain |

*Table 1. Continued on next page*

Table 1. Continued

| Proteins | Sequence information | Notes |
|---|---|---|
| (SH3)<sub>2</sub> | GH**MPAYVKFNYMAEREDELSLIKGTKV IVMEKSSDGWWRGSYNGQVGWFPSNYVTEEGD SPL**SARKASIVKNLKDTLGIGKVKRKPSVPDSASPADD SFVDPGERLYDLN**MPAYVKFNYMAEREDELSL IKGTKVIVMEKSSDGWWRGSYNGQVGWFPSN YVTEEGDSPL**NNPLTSGLEPSPPQCDYIRPSLT GKFAGNPWYYGKVTRHQAEMALNERGHEGDF LIRDSESSPNDFSVSLKAQGKNKHFKVQLKETVYCIG QRKFSTMEELVEHYKKAPIFTSEQGEKLYLVKHLS | Human, two repeats of the second Nck SH3 domain, plus the Nck SH2 domain |
| (SH3)<sub>1</sub> | GH**MPAYVKFNYMAEREDELSLIKGTKVIVME KSSDGWWRGSYNGQVGWFPSNYVTEEGDSPL**NNPLTSGLEPSPPQCDYIRPSLTGKFAGNPWYY GKVTRHQAEMALNERGHEGDFLIRDSESSPNDF SVSLKAQGKNKHFKVQLKETVYCIGQRKFSTMEEL VEHYKKAPIFTSEQGEKLYLVKHLS | Human, one repeat of the second Nck SH3 domain, plus the Nck SH2 domain |
| TIR-1pY | EEHIpYDEVAADPGGSWGGSC | N-terminal rhodamine labeled single pTyr motif from EPEC Tir protein |
| Lck | ANSLEPEPWFFKNLSRKDAERQLLAPGNT HGSFLIRESESTAGSFSLSVRDFDQNQGEVV KHYKIRNLDNGGFYISPRITFPGLHDLVRHYT NASDGLCTKLSRPCQTQKPQKPWWEDEWEVPRE TLKLVERLGAGQFGEVVMGYYNGHTKVAVKSLKQ GSMSPDAFLAEANLMKQLQHPRLVRLYAVVTQEP IYIITEYMENGSLVDFLKTPSGIKLNVNKLL DMAAQIAEGMAFIEEQNYIHRDLRAANILVS DTLSCKIADFGLARLIEDNEYTAREGAKF PIKWTAPEAINYGTFTIKSDVWSFGILLT EIVTHGRIPYPGMTNPEVIQNLERGYRMVRP DNCPEELYHLMMLCWKERPEDRPTFDYLRSVL DDFFTATEGQFQPQP | Human, 119–509, Y505F |

(GE Healthcare, Pittsburgh, PA), evolved with a gradient of 150 → 300 mM NaCl in 20 mM imidazole, pH 8, 1 mM EDTA, and 2 mM DTT, followed by an SD200 column (GE Healthcare) run in 25 mM Hepes, pH 7.5, 150 mM NaCl, 1 mM MgCl<sub>2</sub>, and 2 mM βME. Fractions containing His<sub>8</sub>-Nephrin were concentrated using an Amicon Ultra 3 K concentrator (Millipore, Billerica, MA) and flash frozen in aliquots at −80°C.

Nephrin proteins were phosphorylated at 30°C with 20 nM Lck kinase overnight or with 500 nM Lck for 1 hr. The phosphorylation reaction was quenched with 10 mM EDTA. Kinase and incompletely phosphorylated Nephrin were removed using a source 15 Q column evolved with a gradient of 150 → 250 mM NaCl in 25 mM Hepes, pH 7, and 2 mM βME. The phosphorylated product was further purified using an SD200 column (GE Healthcare) and labeled at its single cysteine residue with maleimide-Alexa 488 fluorophore (Invitrogen, Carlsbad, CA). The labeled protein was separated from unreacted fluorophore using a Source 15 Q column and a Hi-trap desalting column (GE Healthcare). Phosphorylation at one, two, or three sites, for Nephrin1Y, Nephrin2Y, or Nephrin3Y (see *Table 1*), respectively, was confirmed using mass-spectrometry.

GST-Nck and His<sub>6</sub>-N-WASP were expressed in BL21(DE3)T1R cells at 18°C through overnight induction with 1 mM IPTG. Cells expressing GST-Nck were collected by centrifugation and lysed by sonication in 20 mM Tris, pH 8, 200 mM NaCl, 1 mM EDTA, 1 mM DTT, 1 mM PMSF, 1 µg/ml antipain, 1 mM benzamidine, 1 µg/ml leupeptin, and 1 µg/ml pepstatin. The cleared lysate was applied to glutathione sepharose beads (GE) and washed with 10 column volumes of 200 mM NaCl, 20 mM Tris, pH 8, 1 mM DTT, and 1 mM EDTA. The GST tag was removed with TEV protease treatment on the beads at 4°C for 16 hr or at room-temperature for 2 hr. Cleaved Nck was collected by 20 column washes with 20 mM imidazole, pH 7, and 1 mM DTT and applied to a Source 15 Q column using a gradient of 0 → 200 mM NaCl in 20 mM imidazole, pH 7, 1 mM DTT. Fractions containing Nck were pooled, concentrated using an Amicon Ultra 30 K concentrator (Millipore), and passed through a Source 15 S column (GE), using a gradient of 0 → 200 mM NaCl in 20 mM imidazole, pH 7, 1 mM DTT. Fractions containing Nck were concentrated and run through an SD75 column (GE). Pooled fractions were concentrated and flash-frozen

in 25 mM Hepes, pH 7.5, 150 mM NaCl, and 1 mM βME. The $(SH3)_1$, $(SH3)_2$, and $(SH3)_3$ proteins were purified in the same way but excluding the Source 15 S column.

His$_6$-N-WASP expressing cells were collected by centrifugation and lysed by cell disruption (Emulsiflex-C5, Avestin) in 20 mM imidazole, pH 7, 300 mM KCl, 5 mM βME, 0.01% NP-40, 1 mM PMSF, 1 μg/ml antipain, 1 mM benzamidine, and 1 μg/ml leupeptin. The cleared lysate was applied to Ni-NTA agarose (Qiagen), washed with 300 mM KCl, 50 mM imidazole, pH 7, 5 mM βME, and eluted with 100 mM KCl, 300 mM imidazole, pH 7, and 5 mM βME. The elute was further purified over a Source 15 Q column using a gradient of 250 → 450 mM NaCl in 20 mM imidazole, pH 7, and 1 mM DTT. The His$_6$-tag was removed by TEV protease at 4°C for 16 hr or at room-temperature for 2 hr. Cleaved N-WASP was then applied to a Source 15 S column using a gradient of 110 → 410 mM NaCl in 20 mM imidazole, pH 7, 1 mM DTT. Fractions containing N-WASP were concentrated using an Amicon Ultra 10 K concentrator (Millipore), passed through an SD200 column, concentrated and flash-frozen in 25 mM Hepes, pH 7.5, 150 mM NaCl, and 1 mM βME. N-WASP (BPVCA with single cysteine) and Nck (cysteine-modified, see *Table 1*) were labeled with Alexa488/568/647. For labeling purposes, the pure protein after Source15S was desalted into a buffer without reducing agent (25 mM Hepes, pH 7, 150 mM NaCl) and reacted with a maleimide-conjugated fluorophore for 2 hr at room temperature. The reaction was quenched with DTT and the fluorophore was removed using a Source15Q and SD75/Hi-trap desalting columns.

His$_6$-Lck kinase was expressed from baculovirus in *Spodoptera frugiperta* (Sf9) cells. Cells were harvested in 50 mM Tris, pH 7.5, 100 mM NaCl, 5 mM βME and 0.01% NP-40, 1 mM PMSF, 1 μg/ml antipain, 1 mM benzamidine, and 1 μg/ml leupeptin. Cells were lysed by douncing on ice ~10 times. The cleared lysate was applied to Ni-NTA agarose beads equilibrated with 20 mM Tris, pH 7.5, 500 mM NaCl, 20 mM imidazole, 5 mM βME, and 10% glycerol (Buffer A), washed with Buffer A containing 1 M NaCl, and then eluted with Buffer A containing 200 mM imidazole 7.5 and 100 mM NaCl. The elute was applied to a Source 15 Q column using a gradient of 100 → 300 mM NaCl in 25 mM Hepes, pH 7.5, and 2 mM βME. Collected fractions were concentrated (Amicon 10 K, Millipore) and applied to an SD75 column in 25 mM Hepes, pH 7.5, 150 mM NaCl, and 1 mM βME.

## Supported lipid bilayers

Liposomes were prepared as follows. A mixture of 99% DOPC and 1% Ni$^{2+}$-NTA DOGS (Avanti Polar Lipids, Alabaster, Alabama) was dried under argon and further dried under vacuum overnight. The dried mixture was hydrated with MilliQ water for 3 hr. Buffer (25 mM Hepes, pH 7.5, 150 mM NaCl, 1 mM MgCl$_2$) was added to the hydrated multi-lamellar vesicle solution. Small unilamellar vesicles (SUVs) were prepared by 21 passes through an extruder (Avanti) fitted with 80 nm and again seven times with a fresh 80 nm or 30 nm filter. In our hands, changing the filter and re-extruding produced more consistently homogeneous liposomes. SUVs made by this method were stored at 4°C and used within 2 days of extrusion.

To make supported lipid bilayers, chambered glass coverslips (Lab-tek, Cat #155409) were cleaned with 50% isopropanol, washed with Milli-Q water, and then incubated for 2 hr in 6 M NaOH. We found that cleaning the glass and using it within the few hours after cleaning was important to get consistent fluidity of the supported bilayers. Therefore, all experiments were performed within 8 hr of cleaning the glass substrate. After extensive further washes with Milli-Q water, 150 μl of room temperature SUV solution containing 0.5 to 1 mg/ml lipid was added to the coverslips and incubated for 10 min. Unadsorbed vesicles were removed by a three-step wash totaling a 216-fold dilution. BSA, 0.1% (Sigma A3294, protease-free, St. Louis, MO) in 25 mM Hepes, pH 7.5, 150 mM NaCl, 1 mM MgCl$_2$ was used to block the surface for 45 min, yielding a total solution volume of 200 μl. The surface was washed again with 25 mM Hepes, pH 7.5, 150 mM NaCl, 1 mM MgCl$_2$, and 0.1% BSA in two steps totaling a 36-fold dilution. His$_8$-p-Nephrin was added to the bilayer at 100 nM and incubated for 1 hr and washed twice totaling a 36-fold dilution. This procedure yielded 200 μl solution above the bilayer containing 2.8 nM His$_8$-p-Nephrin (assuming a negligible fraction of the total protein binds the bilayer). Subsequent experiments were performed after waiting 30 min to allow the His$_8$ attachment to the bilayer to stabilize (*Figure 2— figure supplement 1D*). Precise control of the timing and dilution-factor of all wash steps was critical to obtaining consistent p-Nephrin densities on the bilayers (quantified as described below). All experiments were performed in 25 mM Hepes, pH 7.5, 150 mM NaCl, 1 mM MgCl$_2$, 1 mM BME, and 0.1% BSA.

## Measurement of nephrin density on supported lipid bilayers

The density of His$_8$-p-Nephrin on the supported lipid bilayers was quantified as previously described (*Galush et al., 2008*; *Salaita et al., 2010*). Briefly, SUVs containing fluorescent lipid (OG-DHPE, Invitrogen)

were made as described above and were used to generate a standard curve of OG-DHPE concentration vs fluorescence intensity on a Nikon Eclipse Ti microscope using a 20× objective focusing deep into the solution and away from the glass (*Figure 2—figure supplement 1A*). The slope of the standard curve was denoted as I-labeled SUV. Using the identical settings, a similar standard curve was made using His$_8$-p-Nephrin-Alexa488 in solution, with slope I-labeled protein (*Figure 2—figure supplement 1B*). I-labeled protein was identical in the presence or absence of Ni-NTA-containing SUVs at 9.5 µM Ni-NTA concentration (minimum of 158-fold excess over His$_8$-p-Nephrin), showing that the His$_8$-p-Nephrin-Alexa488 fluorescence does not change upon binding lipid. The correction factor F, denoted by F = I-labeled protein/I-labeled SUV, represents the intrinsic brightness of and sensitivity of the microscope for His$_8$-p-Nephrin-Alexa488 vs OG-DHPE. Since the OG and Alexa488 fluorophores have very similar excitation and emission spectra, F should be an instrument-independent parameter.

The SUVs containing OG-DHPE were combined in different ratios with non-fluorescent SUVs to make supported bilayers with OG-DHPE densities between 0.05 and 0.4%. Assuming the surface area of the lipid head groups to be 69 Å$^2$ (*Kucerka et al., 2005*), this corresponded to OG-DHPE densities of 1430–11,440 molecules/µm$^2$. A standard curve of bilayer fluorescence intensity on a Nikon Eclipse Ti microscope and a 100× objective vs fluorophore density was then generated from these bilayers. To obtain the density of His$_8$-p-Nephrin-Alexa488 on the supported bilayers, the measured fluorescence intensity was first divided by F, and the result was analyzed with the standard curve of bilayers with OG-DHPE (*Figure 2—figure supplement 1C*). We note that this approach assumes that F is the same on the SLB as when His$_8$-p-Nephrin-Alexa488 and OG-DHPE are associated with SUVs in free solution.

To examine the potential changes in Alexa488 fluorescence as a function of p-Nephrin density, we generated supported bilayers as above with 10–100% Alexa488-labeled p-Nephrin. Intensity remained linear up to ~60% labeling. Initial measurements suggested that the density change in p-bephrin upon clustering is fourfold. Therefore, we used p-Nephrin labeled with 15% or less Alexa488 for all quantitative image analyses.

## Critical concentration measurements

For critical concentration of clustering measurements, images were collected on a Nikon Eclipse Ti microscope equipped with an Andor iXon Ultra 897 EM-CCD camera, with a 100× objective in epi-fluorescence mode. Background was collected with supported bilayers containing non-fluorescent lipids and subtracted from all images before processing. Images were corrected for uneven illumination and detector sensitivity as previously described (*Wu et al., 2008*). Briefly, pixel intensities across a homogeneous bilayer containing p-NephrinA488 were normalized to the maximum intensity of the image to obtain pixel-by-pixel correction factors (in a 0 to 1 range). Experimental images were then corrected by dividing by these factors.

Images were thresholded using the triangle algorithm in Image J. The fractional intensity of the clustered regions was then calculated by dividing the integrated intensity of the thresholded image by that of the non-thresholded image. Analyzing the clusters using the triangle algorithm or the Maximum Entropy algorithm yielded the same critical concentrations. Similar thresholding results were obtained using an iterative manual procedure to identify pixels with intensity greater than three standard deviations above the mean of the non-clustered regions. Thus, our calculation of fractional intensity in the clustered regions and our consequent determination of critical concentration are not dependent on the method used to identify clusters.

## Size distribution and spatial distribution analyses

For the data in *Figure 2B,C*, 512 by 512 pixel images were taken at 93 randomly selected areas of a sample with clusters made using p-NephrinA488, 1 µM Nck, and 1 µM N-WASP. The images were background corrected as described above, flattened using the rolling-ball method in ImageJ, and thresholded using the triangle method. The clusters were binned according to size (excluding those at the image edges) and the distribution was fit to a single-exponential using Graph-pad Prism. The size distributions in *Figure 3B* were determined similarly from single images obtained at each time point.

To analyze the spatial distribution of puncta, each thresholded image was divided into 25 boxes. In each box, the number of clusters was counted twice—excluding and including clusters at the edges. The average number of edge clusters was obtained from the difference in these values, averaged across all boxes in all images. To eliminate overcounting, for each box half of this value was subtracted

from the number of clusters counted including edges. These data were plotted to obtain a frequency histogram using Graph-pad Prism and fit to a Gaussian distribution.

## Fluorescence recovery after photobleaching (FRAP)

FRAP was performed using a Nikon Eclipse Ti microscope equipped with an Andor iXon Ultra EM-CCD camera. A circle of 1-μm diameter was initially photobleached and recovery followed for up to 1000 s. The images were corrected for drift using the Sift-Align plugin in ImageJ (*Schneider et al., 2012*). Background photobleaching was obtained by imaging under the same conditions, excluding the laser illumination used for photobleaching. Background corrected images were normalized to the intensities of the pre-bleached images and fit to either a single or a double-exponential using Graph-pad Prism. F-tests performed in Prism demonstrated that the double-exponential fits are most appropriate (p-values for all experiments were <0.0001, *Table 2*). In the FRAP experiments, a glucose-oxidase scavenger system with trolox was used to reduce photobleaching during the recovery period. $His_8$-p-NephrinA488 dissociation from the membrane was monitored by the decrease in total fluorescence measured in TIRF mode following washes that afforded a final solution concentration of 2.8 nM (see 'Supported lipid bilayers' section above). To limit the effect of photobleaching, the images at each time point were taken at a different area of the bilayer. The data were fit to a single-exponential with time constant of 2080 s.

## Actin assembly assays

Actin and Arp2/3 complex were purified from rabbit muscle and bovine thymus, respectively, using established methods (*Doolittle et al., 2013a*; *Doolittle et al., 2013b*). G-actin (1 μM, 10% rhodamine labeled) was added to p-Nephrin clusters containing 1 μM Nck and 2 μM N-WASP, with or without 10 nM Arp2/3 complex. Images were collected in TIRF mode every 3 min.

For quantitative analysis, images were background corrected and thresholded as described above. In the p-Nephrin clusters, the average intensities of p-Nephrin and rhodamine-actin were measured for times up to 27 min. For each cluster, $t_{1/2}$ represents the time at which the average actin intensity reaches half its maximum value.

## Isothermal titration calorimetry

ITC was performed using a VP-ITC 200 calorimeter (GE Healthcare). Before the experiment, the proteins were dialyzed in the same buffer (25 mM Hepes, pH 7.5, 150 mM NaCl, 1 mM $MgCl_2$, and 2 mM TCEP). Nck at 150 μM in the syringe was titrated to either triply phosphorylated Nephrin or triply phosphorylated TIR. We assumed that all the three sites in Nephrin were of equal affinity. Isotherms were fit well using NITPIC (*Keller et al., 2012*) and Sedphat (*Houtman et al., 2007*), assuming that all three pTyr sites in p-Nephrin have equal affinity for Nck.

## Acknowledgements

We thank Kate-Luby Phelps, Abhijit Bugde, and Karen Rothberg at the UTSW Live Cell Imaging Facility for advice on imaging; Louis Kerr at the Marine Biology Laboratory for his assistance in getting started at MBL; Xiaolei Su, Markus Taylor, Ron Vale, Julia Rumpf, Jack Taunton, and other participants in the MBL/HHMI Summer Institute for advice and discussion regarding imaging and supported lipid bilayers; Jitu Mayor for discussions regarding phase transitions and membrane organization; Khuloud Jaqaman for advice on image analyses; Chad Brautigam and Thomas Scheuermann at the UTSW

**Table 2.** Statistics of fitting for FRAP data

|  | **p-Nephrin** | **Nck (with p-Nephrin)** | **Nck (with p-TIR)** | **N-WASP** |
|---|---|---|---|---|
| Null hypothesis | Single Exp. | Single Exp. | Single Exp. | Single Exp. |
| Alternate hypothesis | Double Exp. | Double Exp. | Double Exp. | Double Exp. |
| p value | <0.0001 | <0.0001 | <0.0001 | <0.0001 |
| Conclusion (alpha = 0.05) | Reject null hypo. | Reject null hypo. | Reject null hypo. | Reject null hypo. |
| Preferred model | Double Exp. | Double Exp. | Double Exp. | Double Exp. |
| F (DFn, DFd) | 64.16 (2282) | 47.33 (2635) | 46.72 (2379) | 48.64 (2379) |

Molecular Biophysics Resource for assistance with ITC; Salman Banani, Jonathan Ditlev, Soyeon Kim, and Shae Padrick for critical reading of the manuscript; and members of the Rosen lab for helpful discussions.

## Additional information

### Funding

| Funder | Grant reference number | Author |
|---|---|---|
| Howard Hughes Medical Institute | HHMI Collaborative Innovations Award (Work performed at MBL/HHMI Summer Institute) | Michael Rosen |
| National Institute of General Medical Sciences | R01-GM56322 | Michael Rosen |
| Welch Foundation | I-1544 | Michael Rosen |
| Chilton Foundation | | Michael Rosen |

The funders had no role in study design, data collection and interpretation, or the decision to submit the work for publication.

### Author contributions

SB, Conception and design, Acquisition of data, Analysis and interpretation of data, Drafting or revising the article; MKR, Conception and design, Analysis and interpretation of data, Drafting or revising the article

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
