## [Decision Letter]

Thank you for sending your work entitled “Phase transitions of multivalent proteins can promote clustering of membrane receptors” for consideration at *eLife*. Your article has been favorably evaluated by John Kuriyan (Senior editor) and 3 reviewers, one of whom is a member of our Board of Reviewing Editors.

The Reviewing editor and the other reviewers discussed their comments before we reached this decision, and the Reviewing editor has assembled the following comments to help you prepare a revised submission.

This paper is a follow-up of the Nature paper where Michael Rosen et al. demonstrated that the cytosolic tail of nephrin, Nck and N-Wasp interacted to form liquid droplets by a phase transition depending on concentration and phosphorylation of the tyrosines in the nephrin tail. Here, a construct of the nephrin cytosolic domain that can be hooked on to the supported bilayers by a lipid tail. This is a nice experimental system where they can follow the reaction dynamics of the liquid droplet formation by different biophysical methods. The data clearly confirm previous conclusion that these four proteins coalesce by a liquid phase separation to form puncta at the bilayer. They also very nicely show how Arp2/3 and actin bind to these assemblies.

The main problem we have right now, apart from the experimental points listed below, is the claim that the coming together of the artificial lipid tails in the membrane is a phase transition. The phase separation is driven by the formation of the liquid droplet and the tails coming together in the bilayer is just a clustering process driven by binding to Nck in the fluid assembly. The process is a consequence of the 3-dimensional phase separation outside the bilayer. Therefore, the data say nothing about the involvement or lack of involvement of the lipids in the bilayer. The experiment is designed in such a way that it cannot be known how the trans-membrane nephrin molecule does its job in the slit diaphragm plasma membrane. We therefore feel that you need to restate the conclusions to justify publication in *eLife*.

More generally, some of the quantification and discussion of phase separation is incorrect or at least imprecise, which must be addressed before publication. The following are the primary concerns:

1) The cluster density is measured in Figure 2, by showing the “given number of clusters within 93 randomly selected 56x56 um regions”. This distribution is Gaussian. What is learned by this? A manuscript is cited (Goswami et.al. 2008) but the analysis in this manuscript seems to be quite different from the one employed here. My concern is that this doesn't really tell you anything; I suspect that the Central Limit Theorem dictates that Gaussian distribution would ‘always’ result from this analysis procedure, independent of the particular spatial distribution.

2) The cluster size distribution is reported to be exponential under different conditions (Figures 2 and 3). There is a fit in 2c, but no fits are provided in 3b to evaluate whether the data are truly exponential. The most convincing way to demonstrate an exponential is to plot the data on a log-lin plot, on which the data will appear as a straight line. The reason why this is important is that the precise form of the distribution is unlikely to be exponential over all times, since the form depends nucleation rate, growth rate, coalescence rate etc. (for an example of a different type of distribution – a power law – seen in intracellular liquid phases, see. Brangwynne et.al. PNAS 2011).

3) The reported Gaussian spatial distribution and exponential size distribution are argued to indicate “a stochastic assembly process that is thermodynamically controlled”. But given the concerns described in points 1 and 2 above, this claim needs to be toned down. Moreover, these are the two most ubiquitous distribution forms, and a wide range of processes can give rise to them.

4) Fluorescence recovery curves are fit to double exponentials. Can these fit equally well to single exponentials? The reason for choosing this functional form should be discussed.

5) The authors argue that “polymerization” underlies phase separation – e.g. the subsection in the Results reads “Phase separation occurs through polymerization of p-Nephrin, Nck, and N-WASP”. What is the reason for using the term polymerization? The data presented shows that valency of Nephrin/Nck is important for the critical concentration for cluster formation, but we don't see how this can be considered a “polymerization” process.

6) In Figure 5, while the fast exponent of recovery was about four fold faster with p-Nephrin compared to that with p-TIR, the slow exponent of recovery changed only modestly between p-TIR and p-Nephrin. What does this mean? Were the FRAP experiments with p-Nephrin and p-TIR carried out under conditions when equal fraction of the total pool of these proteins were in clusters?

7) The figure legends should be revised so that they describe the contents of the figures well, without the readers having to read the Results section. For instance, see Figure 1.

8) The authors state: “Clusters do not form in the absence of Ni-NTA lipids...” Why is Nephrin tail binding the supported bilayer essential for cluster formation? In reference 26, the authors showed the same p-Nephrin, NCK, N-WASP underwent phase separation to form droplets in absence of membrane/supported lipid bilayer.

---

## [Author Response]

*The main problem we have right now, apart from the experimental points listed below, is the claim that the coming together of the artificial lipid tails in the membrane is a phase transition. The phase separation is driven by the formation of the liquid droplet and the tails coming together in the bilayer is just a clustering process driven by binding to Nck in the fluid assembly. The process is a consequence of the 3-dimensional phase separation outside the bilayer. Therefore, the data say nothing about the involvement or lack of involvement of the lipids in the bilayer. The experiment is designed in such a way that it cannot be known how the trans-membrane nephrin molecule does its job in the slit diaphragm plasma membrane. We therefore feel that you need to restate the conclusions to justify publication in* eLife.

The referees appear to have misunderstood two key aspects of our work. We apologize for this confusion, which seems to have two components. First, for exactly the reasons stated above, we did not make any claims about the behavior of the lipids in our study. The phase separation we describe is of the p-Nephrin protein and its ligands, forming a 2D structure adjacent to the membrane. This is, in fact, one of the novel aspects of our work, that proteins alone can phase separate to create 2D structures on the membrane in the absence of lipid phase separation. The coupling between lipid phase separation and protein phase separation is an interesting topic of future research, but it is not addressed here. Second, the phase separation of the nephrin tails that we report here is occurring at Nck concentrations ∼50-fold below those necessary for 3D droplet formation. So the process we are observing is not the formation of 3D droplets in solution, followed by binding of those droplets to membrane-attached p-Nephrin. Rather, the entire process is occurring in the 2D plane of the membrane (or more precisely, just above the membrane). We believe that the lower dimensionality of the membrane-attached system affords the lower critical concentration.

We have two responses to the comment that our work cannot provide insight into how nephrin functions in the slit diaphragm. Most importantly, we have used the Nephrin/Nck/N-WASP system as a model to understand multivalent signaling assemblies in general. This is the greatest value of our work, describing assays, behaviors and properties that will be generalizable to many membrane-attached multivalent signaling systems, much more than providing specific insights into Nephrin function in the kidney. Nevertheless, we do feel that our work does, in fact, inform on some aspects of Nephrin’s biological function. Nephrin clustering in podocytes will be controlled by at least three classes of interactions: extracellular, intramembrane and intracellular. As we describe in the Discussion section, these will all work cooperatively to drive clustering of Nephrin and its various ligands, including both proteins and lipids. Our work describes the intracellular interactions in quantitative detail, providing an initial framework that we will expand in the future to ultimately understand the full-length molecule. For these various reasons we feel our conclusions about phase separation of membrane-attached p-Nephrin are sound, but obviously need to be stated with greater clarity in the manuscript. We have made a number of changes to the Abstract and text to address these points.

*More generally, some of the quantification and discussion of phase separation is incorrect or at least imprecise*, *which must be addressed before publication. The following are the primary concerns:*

*1) The cluster density is measured in*
Figure 2*, by showing the “given number of clusters within 93 randomly selected 56x56 um regions”. This distribution is Gaussian. What is learned by this? A manuscript is cited (Goswami et.al. 2008) but the analysis in this manuscript seems to be quite different from the one employed here. My concern is that this doesn't really tell you anything; I suspect that the Central Limit Theorem dictates that Gaussian distribution would ‘always’ result from this analysis procedure, independent of the particular spatial distribution*.

As we understand it, the Central Limit Theorem is predicated on the assumption of independent events. This informed our interpretation that the Gaussian distribution we observe indicates that the clusters are nucleated randomly and independently. While there was no reason a priori to expect otherwise (barring some cooperativity between nucleation events), we feel that demonstrating this with data has some value. The Goswami manuscript describes an actively driven clustering mechanism based on the dynamics of the actomyosin cytoskeleton. Figure 1, which plot the spatial distribution of fluorescence anisotropy as a proxy for the spatial distribution of clusters of GPI-anchored proteins (anisotropy is proportional to the number of clusters in a given area), shows a distribution that is not Gaussian, thus supporting those authors’ claim that GPI clustering is actively driven. Our comparison here was to emphasize that in our simplified system, clustering is not actively driven. We have revised the text in the Results section to make these points more clearly.

*2) The cluster size distribution is reported to be exponential under different conditions (*Figures 2 and 3*). There is a fit in 2c, but no fits are provided in 3b to evaluate whether the data are truly exponential. The most convincing way to demonstrate an exponential is to plot the data on a log-lin plot, on which the data will appear as a straight line. The reason why this is important is that the precise form of the distribution is unlikely to be exponential over all times, since the form depends nucleation rate, growth rate, coalescence rate etc. (for an example of a different type of distribution* – *a power law* – *seen in intracellular liquid phases, see. Brangwynne et.al. PNAS 2011)*.

We thank the referees for urging us to consider these data in greater depth. In response to these comments we have reevaluated our reported data, and collected new data. In Figure 2 we now show a log-linear plot, which clearly shows exponentially distributed sizes. We have also collected a much larger dataset for the timecourse originally reported in Figure 3 in order to get better statistics. These data also show exponential distributions at times out to 20 minutes after Nck/N-WASP addition (now shown in Figure 3—figure supplement 1). Interestingly however, we have examined the distribution of an additional dataset acquired with a high-density of p-Nephrin on the membrane, and found that it follows a power law (new Figure 3—figure supplement 2). Thus, as the referee suggests, the functional form of the size distribution can show at least two different behaviors, due to the variable contributions of nucleation/growth/coalescence rates to the clustering dynamics. Because of these complexities, we have refrained from interpreting these data, beyond demonstrating dynamics of the clusters. A full mechanistic description of the clustering dynamics would require a great deal more experimental data and also development of theory, and is beyond the scope of the present work.

*3) The reported Gaussian spatial distribution and exponential size distribution are argued to indicate “a stochastic assembly process that is thermodynamically controlled”. But given the concerns described in points 1 and 2 above, this claim needs to be toned down. Moreover, these are the two most ubiquitous distribution forms, and a wide range of processes can give rise to them*.

We agree with the referees on this point and have changed our wording to be more cautious in our interpretations of the cluster distributions.

*4) Fluorescence recovery curves are fit to double exponentials. Can these fit equally well to single exponentials? The reason for choosing this functional form should be discussed*.

We tried to fit the data to single exponentials, and found that the fitting is statistically poorer than a double exponential based on the F-test. These statistics were included in early drafts of our manuscript, but were removed during editing. We have returned them to the methods section of the revised manuscript, and direct readers to them in the Results section. In the absence of a mechanistic model for the cluster dynamics, we chose the exponential form as a simple means to identify different kinetic processes in the recovery.

*5) The authors argue that “polymerization” underlies phase separation – e.g. the subsection in the Results reads “Phase separation occurs through polymerization of p-Nephrin, Nck, and N-WASP”. What is the reason for using the term polymerization? The data presented shows that valency of Nephrin/Nck is important for the critical concentration for cluster formation, but we don't see how this can be considered a “polymerization” process*.

Our findings that the system shows a sharp critical concentration, that valency and monomer-monomer binding affinity strongly affect the critical concentration, that the structures formed are macroscopic in size, and that the structures show complex dynamics which are also dependent on monomer-monomer affinity (likely on the monomer-monomer dissociation rate) all suggest that the system is undergoing a sol-gel transition. Such transitions involve the formation of macroscopic polymers connected by module-module interactions. The dissolution of the clusters by a monovalent competitor is also consistent with a polymeric nature. While we agree that our data do not constitute proof that the clusters consist of p-Nephrin/Nck/N-WASP polymers, we feel that the data are sufficiently strong that this is the best description of the molecular nature of the system. In this case, we believe that making an explicit connection to polymer science is important, since our experiments nicely follow a large body of work in that area.

*6) In*
Figure 5*, while the fast exponent of recovery was about four fold faster with p-Nephrin compared to that with p-TIR, the slow exponent of recovery changed only modestly between p-TIR and p-Nephrin*. *What does this mean? Were the FRAP experiments with p-Nephrin and p-TIR carried out under conditions when equal fraction of the total pool of these proteins were in clusters?*

While we do not feel we have enough data to mechanistically characterize the dynamics of the clusters at this stage, a reasonable interpretation of the FRAP recovery data is that the fast component arises from dissociation of Nck and/or its complexes with N-WASP from p-Tir/p-Nephrin (which should be affected by the rate of SH2-pTyr dissociation, and thus SH2-pTyr affinity), while the slow component reflects movement of large clusters in the plane of the membrane (which may be of similar size in the two cases). This model is sufficiently speculative, however, that we have elected to leave it out of the manuscript. Further, as above, our intent in these experiments was not to develop a mechanistic model of the dynamics of the system. That would require an entire paper(s) in its own right. Rather, we aimed to examine whether aspects of the dynamics were dependent on the monomer-monomer binding affinities, as would be expected in a polymeric system. Regarding the last point, yes, in both sets of FRAP experiments, 26 % of p-Tir/p-Nephrin molecules were in the clusters.

*7) The figure legends should be revised so that they describe the contents of the figures well, without the readers having to read the Results section. For instance, see*
Figure 1.

We have revised the figure legends so that they stand alone better.

*8) The authors state: “Clusters do not form in the absence of Ni-NTA lipids...” Why is Nephrin tail binding the supported bilayer essential for cluster formation? In reference 26, the authors showed the same p-Nephrin, NCK, N-WASP underwent phase separation to form droplets in absence of membrane/supported lipid bilayer*.

This relates to the same issue that was raised in the first paragraph, which seems to arise from a misunderstanding regarding our data. The clusters we report here are formed at ∼50-fold lower Nck concentration than needed to form phase separated droplets in solution (i.e. in reference 26). Thus, our observations do not represent binding to membranes of 3D phase separated structures, but rather a phase separation process that only occurs in a 2D plane at the membrane. The much lower critical concentration likely is due to the decreased dimensionality of the system when p-Nephrin is attached to the bilayer.